# SAFE COLLABORATIVE FILTERING

**Riku Togashi**[a,*] **Tatsushi Oka**[b] **Naoto Ohsaka**[a] **Tetsuro Morimura**[a]

[a] CyberAgent
`rtogashi@acm.org`[*]`,{ohsaka_naoto,morimura_tetsuro}@cyberagent.co.jp`
[b] Department of Economics, Keio University
`tatsushi.oka@keio.jp`

## ABSTRACT

Excellent tail performance is crucial for modern machine learning tasks, such as algorithmic fairness, class imbalance, and risk-sensitive decision making, as it ensures the effective handling of challenging samples within a dataset. Tail performance is also a vital determinant of success for personalized recommender systems to reduce the risk of losing users with low satisfaction. This study introduces a *"safe"* collaborative filtering method that prioritizes recommendation quality for less-satisfied users rather than focusing on the average performance. Our approach minimizes the conditional value at risk (CVaR), which represents the average risk over the tails of users' loss. To overcome computational challenges for web-scale recommender systems, we develop a robust yet practical algorithm that extends the most scalable method, implicit alternating least squares (iALS). Empirical evaluation on real-world datasets demonstrates the excellent tail performance of our approach while maintaining competitive computational efficiency.

## 1 INTRODUCTION

Owing to the widespread implementation of collaborative filtering (CF) techniques (Hu et al., 2008; Koren et al., 2009; Rendle et al., 2022; Steck, 2019a), recommender systems have become ubiquitous in web applications and are increasingly impacting business profits. The quality of personalization is critical, particularly for users with low satisfaction, as they are essential for driving user growth, yet their disengagement poses a serious risk to the survival of the business. This is where the problem lies with conventional methods based on empirical risk minimization (ERM), which focus on performance in terms of *averages* with no regard given to harmful errors for the tail users. In this paper, we address this problem by utilizing a risk measure, conditional value at risk (CVaR) (Pflug, 2000; Rockafellar and Uryasev, 2000; 2002), which represents the average risk for tail users.

Thus far, significant research efforts have been devoted to the computational aspects of risk-averse optimization (Alexander et al., 2006; Xu and Zhang, 2009). We are also interested in the scalability of CVaR minimization for web-scale recommender systems, which must meet strict computational requirements. The challenge of risk-averse CF is scalability in *two dimensions*: users and items. In the context of personalized ranking, the focus has been on the scalability for the *items* to be ranked. Numerous studies (Rendle et al., 2009; Weimer et al., 2007; 2008a;b) have addressed the intractability of direct ranking optimization via the "score-and-sort" approach, where, given a user, a method predicts scores for all items and then sorts them according to their scores. To attain further scalability for a large item catalogue, practical methods adopt pointwise loss functions, enabling parallel optimization for users and items (Bayer et al., 2017; He et al., 2016; Hu et al., 2008). The key to these efficient methods is the *separability* of pointwise objectives, which allows block coordinate algorithms, such as alternating least squares (ALS). However, integrating CVaR minimization into this approach faces a severe obstacle because of the non-smooth and non-linear functions (i.e., check functions), which hinder the separability and parallel computation for items.

Our contribution in this paper is to devise a practical algorithm for risk-averse CF by eliminating the mismatch between CVaR minimization and personalized ranking. The paper is structured as follows. In Section 2, we show that applying CVaR minimization to matrix factorization (MF) is

---

[*]Corresponding author

computationally expensive. This is mainly due to the check function even with a block multi-convex, separable loss. In Section 3, we overcome this challenge by using a smoothing technique of quantile regression (Fernandes et al., 2021; He et al., 2021; Tan et al., 2022) and establish a block coordinate solver that is embarrassingly parallelizable for both users and items. The related work is discussed in Section 4, and our experiments are described in Section 5.

## 2 SETTING AND CHALLENGE

Following conventional studies (Weimer et al., 2007; 2008a;b), we view collaborative filtering as a ranking problem. In this setting, we have access to the set $\mathcal{S}$ of implicit feedback $(i, j)$ indicating that user $i$ has preferred item $j$. We denote the sets of users and items observed in $\mathcal{S}$ as $\mathcal{U}$ and $\mathcal{V}$, respectively. For convenience, we also define $\mathcal{V}_i$ to be the set of items preferred by user $i$, i.e., $\mathcal{V}_i := \{j \in \mathcal{V} \mid (i, j) \in \mathcal{S}\}$, and $\mathcal{U}_j$ to be the set of users that have selected item $j$, i.e., $\mathcal{U}_j := \{i \in \mathcal{U} \mid (i, j) \in \mathcal{S}\}$. The aim of this problem is to learn a scoring function $\boldsymbol{f}_\theta$ parameterized by $\theta$, which produces a structured prediction $\boldsymbol{f}_\theta(i) \in \mathbb{R}^{|\mathcal{V}|}$ having the same order of underlying $i$'s preferences on $\mathcal{V}$; we may denote the predicted score for $(i, j)$ by $f_\theta(i, j)$.

**Pairwise ranking optimization.** Conventional studies (Lee et al., 2014; Park et al., 2015; Rendle et al., 2009) often formulate this problem as pairwise ranking optimization within the ERM framework, aiming to minimize a population risk $\min_\theta \mathbb{E}_{(i, \mathcal{V}_i)} [\ell_{\text{pair}}(\boldsymbol{f}_\theta(i), \mathcal{V}_i)]$, where $\mathbb{E}_{(i, \mathcal{V}_i)}$ represents the expectation over user $i$ with feedback $\mathcal{V}_i$. Here, the loss function $\ell_{\text{pair}}$ is expressed as follows:

$$\ell_{\text{pair}}(\boldsymbol{f}_\theta(i), \mathcal{V}_i) := \frac{1}{|\mathcal{V}_i|} \sum_{j \in \mathcal{V}_i} \sum_{j' \in \mathcal{V}} \mathbb{1}\{f_\theta(i, j') \geq f_\theta(i, j)\}, \tag{1}$$

where $\mathbb{1}\{\cdot\}$ is the indicator function. Because this loss function is piece-wise constant and intractable, its convex upper bounds are used, such as margin hinge loss (Weimer et al., 2008b) and softplus loss (Rendle et al., 2009). However, such loss functions lead to non-separability for items; that is, a loss cannot be decomposed into the sum of independent functions $\{g_j\}_{j \in \mathcal{V}}$ for items, $\ell_{\text{pair}}(\boldsymbol{f}_\theta(i), \mathcal{V}_i) \neq \sum_{j \in \mathcal{V}} g_j(f_\theta(i, j))$. Thus, this approach often relies on gradient-based optimizers with negative sampling techniques (Rendle and Freudenthaler, 2014), which however suffer from slow convergence (Chen et al., 2023; Yu et al., 2014).

**Convex and separable upper bound.** To address this non-separability issue, conventional methods utilize convex and separable loss functions, i.e., pointwise loss (Hu et al., 2008; Zhou et al., 2008). We also adopt the following convex and separable upper bound of Eq. (1),

$$\frac{1}{|\mathcal{V}_i|} \sum_{j \in \mathcal{V}_i} [1 - f_\theta(i, j)]^2 + \beta_0 \cdot \sum_{j \in \mathcal{V}} f_\theta(i, j)^2, \tag{2}$$

where $\beta_0 \geq 1/|\mathcal{V}|$ is a hyperparameter. The derivation is deferred to Appendix A.

We here implement the model of $\boldsymbol{f}_\theta$ using matrix factorization (MF), which is a popular model owing to its scalability (Hu et al., 2008; Koren et al., 2009; Rendle et al., 2022). The model parameters of an MF comprise two blocks, i.e., $\theta = (\mathbf{U}, \mathbf{V})$ where $\mathbf{U} \in \mathbb{R}^{|\mathcal{U}| \times d}$ and $\mathbf{V} \in \mathbb{R}^{|\mathcal{V}| \times d}$ are the embedding matrices of users and items, respectively. The prediction for user $i$ is then defined by $\boldsymbol{f}_\theta(i) = \mathbf{V}\mathbf{u}_i$, where $\mathbf{u}_i \in \mathbb{R}^d$ represents the $i$-th row of $\mathbf{U}$, and the ERM objective can be expressed as follows:

$$\min_{\mathbf{U}, \mathbf{V}} \frac{1}{|\mathcal{U}|} \sum_{i \in \mathcal{U}} \ell(\mathbf{V}\mathbf{u}_i, \mathcal{V}_i) + \Omega(\mathbf{U}, \mathbf{V}), \tag{3}$$

where

$$\ell(\mathbf{V}\mathbf{u}_i, \mathcal{V}_i) := \frac{1}{|\mathcal{V}_i|} \sum_{j \in \mathcal{V}_i} \frac{1}{2}(1 - \mathbf{u}_i^\top \mathbf{v}_j)^2 + \frac{\beta_0}{2} \|\mathbf{V}\mathbf{u}_i\|_2^2. \tag{4}$$

Here, $\Omega(\mathbf{U}, \mathbf{V}) = \frac{1}{2}\|\mathbf{\Lambda}_u^{1/2}\mathbf{U}\|_F^2 + \frac{1}{2}\|\mathbf{\Lambda}_v^{1/2}\mathbf{V}\|_F^2$ denotes L2 regularization, with diagonal matrices $\mathbf{\Lambda}_u \in \mathbb{R}^{|\mathcal{U}| \times |\mathcal{U}|}$ and $\mathbf{\Lambda}_v \in \mathbb{R}^{|\mathcal{V}| \times |\mathcal{V}|}$ representing user- and item-dependent Tikhonov weights. Because a pointwise loss enables a highly scalable ALS algorithm owing to separability, it is widely used for scalable methods, such as implicit alternating least squares (iALS) (Hu et al., 2008). In fact,

iALS is implemented in practical applications (Meng et al., 2016) and is still known as one of the state-of-the-art methods (Rendle et al., 2022). The only difference between our loss and iALS loss is the normalization factor $1/|\mathcal{V}_i|$ in the first term. For further details, see Appendix B.

**Conditional value at risk (CVaR).**   In contrast to ERM, which corresponds to the *average* case, we focus on a more pessimistic scenario. Specifically, the objective of $(1 - \alpha)$-CVaR minimization involves a conditional expectation,

$$\min_{\theta} \mathbb{E}_{(i,\mathcal{V}_i)} \left[ \ell(\boldsymbol{f}_\theta(i), \mathcal{V}_i) \mid \ell(\boldsymbol{f}_\theta(i), \mathcal{V}_i) \geq \ell_{1-\alpha} \right],$$

where $\ell_{1-\alpha}$ is the $(1 - \alpha)$-quantile of the loss distribution, called *value at risk (VaR)* (Jorion, 2007). Since this is difficult to optimize, Rockafellar and Uryasev (2000; 2002) proposed the following reformulation:

$$\min_{\theta,\xi} \left\{ \xi + \alpha^{-1} \mathbb{E}_{(i,\mathcal{V}_i) \sim \mathbb{P}} [\max(0, \ell(\boldsymbol{f}_\theta(i), \mathcal{V}_i) - \xi)] \right\}.$$

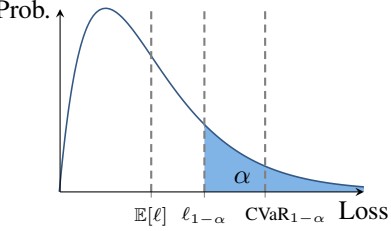

Figure 1: Illustration of CVaR

This objective is rather easy to optimize, as it is block multi-convex w.r.t. both $\xi$ and $\theta$ when the loss function is convex w.r.t. $\theta$.

**CVaR-MF.**   Now, we apply the above CVaR formulation to Eq. (3) as follows:

$$\min_{\mathbf{U},\mathbf{V},\xi} \left\{ \xi + \frac{1}{\alpha|\mathcal{U}|} \sum_{i \in \mathcal{U}} \max(0, \ell(\mathbf{V}\mathbf{u}_i, \mathcal{V}_i) - \xi) + \Omega(\mathbf{U}, \mathbf{V}) \right\}. \tag{5}$$

However, optimizing this objective is still challenging. Although subgradient methods (Rockafellar and Uryasev, 2000) enable parallel optimization of non-smooth objectives without requiring separability, such first-order solvers are often impractical due to slow convergence. On the other hand, the obstacle in designing an efficient algorithm lies in the non-linearity of the ramp function $\rho_1(x) = \max(0, x)$. It destroys the objective's separability w.r.t. items, making it impossible to solve the subproblem for each row of $\mathbf{V}$ in parallel, even with the separable upper bound in Eq. (4).

# 3   SAFER$_2$: SMOOTHING APPROACH FOR EFFICIENT RISK-AVERSE RECOMMENDER

## 3.1   CONVOLUTION-TYPE SMOOTHING FOR CVAR

To overcome the non-smoothness of CVaR, we utilize a smoothing technique for quantile regression (Fernandes et al., 2021; He et al., 2021; Man et al., 2022; Tan et al., 2022), which involves the integral convolution with smooth functions, called mollifiers, as studied by Friedrichs (1944); Schwartz (1951); Sobolev (1938) among others. We introduce the notation $k_h(\cdot) := h^{-1}k(\cdot/h)$, where $k(\cdot)$ is a kernel density function satisfying $\int k(u)du = 1$ and $h > 0$ denotes bandwidth, while $K_h$ represents its CDF; $K_h(u) := \int_{-\infty}^{u} k_h(v)dv$[1]. We then define a smoothed check function $\rho_{1-\alpha} * k_h \colon \mathbb{R} \to \mathbb{R}$ as follows:

Figure 2: Convolution-type smoothing.

$$(\rho_{1-\alpha} * k_h)(u) := \int \rho_{1-\alpha}(v)k_h(v - u)dv, \tag{6}$$

where $*$ is called the convolution operator. The resulting function $\rho_{1-\alpha} * k_h$ attains strict convexity when $k_h$ is strictly convex[2]. We then obtain the objective of convolution-type smoothed CVaR:

$$\min_{\mathbf{U},\mathbf{V},\xi} \Big\{ \xi + \underbrace{\frac{1}{\alpha|\mathcal{U}|} \sum_{i \in \mathcal{U}} (\rho_1 * k_h)(\ell(\mathbf{V}\mathbf{u}_i, \mathcal{V}_i) - \xi)}_{\Psi_{1-\alpha}(\mathbf{U},\mathbf{V},\xi)} + \Omega(\mathbf{U}, \mathbf{V}) \Big\}. \tag{7}$$

---

[1]We discuss the implementation of $k_h$ in Appendix D.

[2]The properties of smoothed check functions are discussed in Appendix C.4

We hereafter denote the functional $\Psi_{1-\alpha}(\mathbf{U}, \mathbf{V}, \xi)$ as $(1-\alpha)$-CtS-CVaR. If $\ell$ is block multi-convex w.r.t. $\mathbf{U}$ and $\mathbf{V}$, then $\Psi_{1-\alpha}(\mathbf{U}, \mathbf{V}, \xi)$ is also block multi-convex w.r.t. $\mathbf{U}$, $\mathbf{V}$, and $\xi$ because $(\rho_1 * k_h)$ is convex and non-decreasing. Therefore, we consider a block coordinate algorithm, where we cyclically update each block while fixing the other blocks to the current estimates. That is,

$$\xi^{(t+1)} = \arg\min_{\xi} \Psi_{1-\alpha}(\mathbf{U}^{(t)}, \mathbf{V}^{(t)}, \xi), \tag{8}$$

$$(\mathbf{U}^{(t+1)}, \mathbf{V}^{(t+1)}) = \arg\min_{\mathbf{U}, \mathbf{V}} \Psi_{1-\alpha}(\mathbf{U}, \mathbf{V}, \xi^{(t+1)}). \tag{9}$$

Because the subproblems lack a closed-form update and block separability, we develop our proposed method, *Smoothing Approach For Efficient Risk-averse Recommender* (SAFER$_2$), which exploits the objective's smoothness to yield block multi-convex and separable optimization. In the following, we present the overall algorithm of SAFER$_2$, and Appendix D contains the detailed implementation.

## 3.2 CONVOLUTION-TYPE SMOOTHED QUANTILE ESTIMATION

The smoothed $(\rho_{1-\alpha} * k_h)$ is twice differentiable, so a natural approach to finding the solution $\xi^*$ is to use the Newton–Raphson (NR) algorithm. At iteration $l < L$ in the $(k+1)$-th update, it estimates $\xi$ as follows:

$$\xi_{l+1}^{(t+1)} = \xi_l^{(t+1)} - \gamma_l \cdot d_l^{(t+1)}, \quad \text{where } d_l^{(t+1)} := \frac{\nabla_\xi \Psi_{1-\alpha}(\mathbf{U}^{(t)}, \mathbf{V}^{(t)}, \xi_l^{(t+1)})}{\nabla_\xi^2 \Psi_{1-\alpha}(\mathbf{U}^{(t)}, \mathbf{V}^{(t)}, \xi_l^{(t+1)})}, \tag{10}$$

where $\gamma_l > 0$ represents the step size. The convolution-type smoothing was originally used with check functions for quantile regression, and we are the first to apply it to smooth the CVaR objective. Interestingly, even if we apply the convolution-type smoothing to $\max(0, \cdot)$ in CVaR, optimizing $\xi$ for a given $\mathbf{U}$ and $\mathbf{V}$ is equivalent to a smoothed (unconditional) quantile estimation; the proof is deferred to Appendix C.3.

**Efficient loss computation.** The naive computation of $\ell(\mathbf{Vu}_i, \mathcal{V}_i)$ for every user is infeasible in large-scale settings owing to the score penalty $\|\mathbf{Vu}_i\|_2^2$, which implies the materialization of the recovered matrix $\mathbf{UV}^\top$ with the cost of $\mathcal{O}(|\mathcal{U}||\mathcal{V}|d^2)$. We can reduce this cost by pre-computing and caching the Gram matrix $\mathbf{V}^\top\mathbf{V} \in \mathbb{R}^{d \times d}$ in $\mathcal{O}(|\mathcal{V}|d^2)$ and computing the loss of user $i$ with $\|\mathbf{Vu}_i\|_2^2 = \mathbf{u}_i^\top (\mathbf{V}^\top\mathbf{V})\mathbf{u}_i$ in $\mathcal{O}(d^2)$, thus leading to the overall cost of $\mathcal{O}((|\mathcal{U}| + |\mathcal{V}|)d^2)$.

**Stochastic optimization for many users.** Estimating $\xi$ can be expensive because of the $\mathcal{O}(|\mathcal{U}|)$ computational cost for each NR step, particularly for many users. We can alleviate this extra cost by using stochastic algorithms (Roosta-Khorasani and Mahoney, 2019; Xu et al., 2016), in which we randomly sample the user subset $\mathcal{U}_b \subset \mathcal{U}$ and then estimate the direction by a sample average approximation $\widehat{\Psi}_{1-\alpha}(\mathbf{U}, \mathbf{V}, \xi) := \xi + (\alpha|\mathcal{U}_b|)^{-1} \sum_{i \in \mathcal{U}_b} (\rho_1 * k_h)(\ell(\mathbf{Vu}_i, \mathcal{V}_i) - \xi)$. This also allows us to use a backtracking line search (Armijo, 1966) to determine an appropriate step size $\gamma_t$ while keeping computational costs in $\mathcal{O}(|\mathcal{U}_b|L)$ where $L > 0$ is the number of NR iterations.

## 3.3 PRIMAL-DUAL SPLITTING FOR PARALLEL COMPUTATION

The major obstacle to the scalability for items is the row-wise coupling of $\mathbf{V}$ stemming from non-linear composite $(\rho_1 * k_h)(\ell(\mathbf{Vu}_i, \mathcal{V}_i))$. Because $(\rho_1 * k_h)$ is closed and convex, we can express it as its biconjugate $(\rho_1 * k_h)(r) = \max_z \{z \cdot r - (\rho_1 * k_h)^*(z)\}$. This allows us to reformulate the original subproblem of $\mathbf{U}$ and $\mathbf{V}$ as the following saddle-point optimization:

$$\min_{\mathbf{U}, \mathbf{V}} \max_{\mathbf{z}} \left\{ \xi + \frac{1}{\alpha|\mathcal{U}|} \sum_{i \in \mathcal{U}} [z_i \cdot (\ell(\mathbf{Vu}_i, \mathcal{V}_i) - \xi) - (\rho_1 * k_h)^*(z_i)] + \Omega(\mathbf{U}, \mathbf{V}) \right\}, \tag{11}$$

where $\mathbf{z} \in \mathbb{R}^{|\mathcal{U}|}$ represents the vector of dual variables. Our algorithm alternatingly updates each block of variables (i.e., $\mathbf{z}$, $\mathbf{U}$ and $\mathbf{V}$) by solving its subproblem. As we will discuss below, owing to convolution-type smoothing, the inner maximization for the dual variable $\mathbf{z}$ can be solved exactly and efficiently, and thus our algorithm operates similarly to a block coordinate descent (Tseng, 2001) for the primal variables $\mathbf{U}$ and $\mathbf{V}$. Through numerical experiments, we will show this algorithm converges in practice when given a sufficiently large bandwidth $h$.

**Conjugate-free optimization.** In the $\mathbf{z}$ step, we solve the problem for $z_i$ of each user $i$,

$$z_i^{(t+1)} = \arg\max_{z_i}\{z_i \cdot (\ell(\mathbf{V}^{(t)}\mathbf{u}_i^{(t)}, \mathcal{V}_i) - \xi^{(t+1)}) - (\rho_1 * k_h)^*(z_i)\},$$

where the convex conjugate $(\rho_1 * k_h)^*$ does not necessarily have a closed-form expression. Fortunately, with convolution-type smoothing, we can sidestep this issue and solve each subproblem without explicitly computing $(\rho_1 * k_h)^*$. Let $r_i^{(t)} := \ell(\mathbf{V}^{(t)}\mathbf{u}_i^{(t)}, \mathcal{V}_i) - \xi^{(t+1)}$ denote the residual for user $i \in \mathcal{U}$. Then, the first-order optimality condition for $z_i$ is

$$r_i^{(t)} - \nabla_{z_i}(\rho_1 * k_h)^*(z_i^{(t+1)}) = 0. \tag{12}$$

Exploiting the property of convex conjugate, the gradient $\nabla_{z_i}(\rho_1 * k_h)^*(z_i)$ can be obtained by

$$\nabla_{z_i}(\rho_1 * k_h)^*(z_i) = \arg\min_x \{-x \cdot z_i + (\rho_1 * k_h)(x)\}$$

$$\implies -z_i + 1 - K_h(-\nabla_{z_i}(\rho_1 * k_h)^*(z_i)) = 0 \quad (\because \nabla_x(\rho_1 * k_h)(x) = 1 - K_h(-x))^3$$

$$\iff \nabla_{z_i}(\rho_1 * k_h)^*(z_i) = -K_h^{-1}(1 - z_i). \tag{13}$$

Recall that $K_h$ is the CDF of $k_h$ and therefore has its inverse $K_h^{-1}$ (i.e., the quantile function). Consequently, by using Eqs. (12) and (13), we obtain the closed-form solution of $z_i^{(t+1)}$ as:

$$r_i^{(t)} - \nabla_{z_i}(\rho_1 * k_h)^*(z_i^{(t+1)}) = 0 \iff z_i^{(t+1)} = 1 - K_h(-r_i^{(t)}). \tag{14}$$

Once we have $\xi^{(t+1)}$, all $z_i^{(t+1)}$ can be computed in parallel for users, and $K_h$ can also be computed effortlessly when using standard kernels for $k_h$, such as Gaussian, sigmoid, and logistic kernels[4].

**Re-weighted alternating least squares.** When given dual variables $\mathbf{z}$, the optimization problem of $\mathbf{U}$ and $\mathbf{V}$ forms a re-weighted ERM, i.e., $\min_{\mathbf{U},\mathbf{V}} \left\{ (\alpha|\mathcal{U}|)^{-1} \sum_{i \in \mathcal{U}} z_i \cdot \ell(\mathbf{V}\mathbf{u}_i, \mathcal{V}_i) + \Omega(\mathbf{U}, \mathbf{V}) \right\}$, which is separable and block strongly multi-convex w.r.t. the rows of $\mathbf{U}$ and $\mathbf{V}$. Consequently, this step is efficient to the same extent as the most efficient ALS solver of Hu et al. (2008). It is also worth noting that, as proposed in Hu et al. (2008), we pre-compute Gram matrices $\mathbf{V}^\top\mathbf{V}$ and $\mathbf{U}^\top \text{diagMat}(\mathbf{z})\mathbf{U}$ to reuse them for efficiently solve the separable subproblems in parallel (a.k.a. the Gramian trick (Krichene et al., 2018; Rendle et al., 2022)). It is also possible to parallelize each pre-computation step by considering $\mathbf{V}^\top\mathbf{V} = \sum_{j \in \mathcal{V}} \mathbf{v}_j\mathbf{v}_j^\top$ and $\mathbf{U}^\top \text{diagMat}(\mathbf{z})\mathbf{U} = \sum_{i \in \mathcal{U}} z_i \cdot \mathbf{u}_i\mathbf{u}_i^\top$.

**Tikhonov regularization.** Previous observations have shown that Tikhonov weights $\mathbf{\Lambda}_u$ and $\mathbf{\Lambda}_v$ are critical in enhancing the ranking quality of MF (Rendle et al., 2022; Zhou et al., 2008). We hence develop a regularization strategy for SAFER$_2$ based on *condition numbers*, which characterize the numerical stability of ridge-type problems. Let $\lambda_u^{(i)}$ and $\lambda_v^{(j)}$ denote the $(i, i)$-element of $\mathbf{\Lambda}_u$ and the $(j, j)$-element of $\mathbf{\Lambda}_v$, respectively. We propose the weights for user $i$ and item $j$ as follows:

$$\lambda_u^{(i)} = \frac{\lambda}{\alpha|\mathcal{U}|}\left(1 + \beta_0|\mathcal{V}|\right), \quad \lambda_v^{(j)} = \frac{\lambda}{\alpha|\mathcal{U}|}\left(\sum_{i \in \mathcal{U}_j} \frac{1}{|\mathcal{V}_i|} + \beta_0\alpha|\mathcal{U}|\right). \tag{15}$$

Here, $\lambda > 0$ represents the base weight that requires tuning. This strategy introduces only one hyperparameter and empirically improves final performance. See Appendix E for a detailed derivation.

### 3.4 COMPUTATIONAL COMPLEXITY AND SCALABILITY.

The computational cost of the $\xi$ step is $\mathcal{O}(|\mathcal{U}_b|L)$, and for the $\mathbf{U}$ and $\mathbf{V}$ steps is $\mathcal{O}(|\mathcal{S}|d^2 + (|\mathcal{U}| + |\mathcal{V}|)d^3)$, which is identical to that of iALS (Hu et al., 2008). Therefore, the overall complexity per epoch is $\mathcal{O}(|\mathcal{U}_b|L + |\mathcal{S}|d^2 + (|\mathcal{U}| + |\mathcal{V}|)d^3)$. The linear dependency on $|\mathcal{U}|$ and $|\mathcal{V}|$ can be alleviated arbitrarily by increasing the parallel degree owing to separability. The cubic dependency on $d$ can be circumvented by leveraging the conjugate gradient method (Tan et al., 2016) and subspace-based block coordinate descent (Rendle et al., 2021). The sketch of the algorithm is shown in Algorithm 1. In Appendix D.4, we also discuss a variant of SAFER$_2$ to alleviate the $d^3$ factor.

---

[3]The derivatives of smoothed check functions are provided in Appendix C.1.

[4]For explicit expressions with some kernels, see Remark 3.1 of He et al. (2021).

**Algorithm 1:** SAFER$_2$ solver.

**Input** : $\{\mathcal{V}_i\}_{i \in \mathcal{U}}$
**Output** : $\mathbf{U}, \mathbf{V}$

$(\mathbf{U}, \mathbf{V}, \xi, \{\ell_i\}_{i \in \mathcal{U}}, \mathbf{G}_V) \leftarrow \texttt{Init}(\{\mathcal{V}_i\}_{i \in \mathcal{U}})$
**for** $t \leftarrow 1$ **to** $T$ **do**
  $\xi \leftarrow \texttt{ComputeXi}(\xi, \{\ell_i\}_{i \in \mathcal{U}})$
  **for** $i \leftarrow 1$ **to** $|\mathcal{U}|$ **do**
    $z_i \leftarrow 1 - K_h(\xi - \ell_i)$
    $\mathbf{u}_i \leftarrow \arg\min_{\mathbf{u}}\{\frac{z_i}{\alpha|\mathcal{U}|}\ell(\mathbf{V}\mathbf{u}, \mathcal{V}_i) + \frac{\lambda_u^{(i)}}{2}\|\mathbf{u}\|_2^2\}$
  $\widetilde{\mathbf{G}}_U \leftarrow \sum_{i \in \mathcal{U}} z_i \cdot \mathbf{u}_i \mathbf{u}_i^\top$
  **for** $j \leftarrow 1$ **to** $|\mathcal{V}|$ **do**
    $\mathbf{v}_j \leftarrow \arg\min_{\mathbf{v}}\{\Psi_{1-\alpha}(\mathbf{U}, \mathbf{V}, \xi) + \frac{\lambda_v^{(j)}}{2}\|\mathbf{v}\|_2^2\}$
  $\mathbf{G}_V \leftarrow \sum_{j \in \mathcal{V}} \mathbf{v}_j \mathbf{v}_j^\top$
  $\forall i \in \mathcal{U}, \ell_i \leftarrow \ell(\mathbf{V}\mathbf{u}_i, \mathcal{V}_i)$
**return** $\mathbf{U}, \mathbf{V}$

**Algorithm 2:** Subroutines for SAFER$_2$.

**Function** $\texttt{Init}(\{\mathcal{V}_i\}_{i \in \mathcal{U}})$:
  $\forall i \in \mathcal{U}, \mathbf{u}_i \sim \mathcal{N}(\vec{0}, (\sigma/\sqrt{d})\mathbf{I}_d)$
  $\forall j \in \mathcal{V}, \mathbf{v}_j \sim \mathcal{N}(\vec{0}, (\sigma/\sqrt{d})\mathbf{I}_d)$
  $\mathbf{G}_V \leftarrow \sum_{j \in \mathcal{V}} \mathbf{v}_j \mathbf{v}_j^\top$
  $\forall i \in \mathcal{U}, \ell_i \leftarrow \ell(\mathbf{V}\mathbf{u}_i, \mathcal{V}_i)$
  $\xi \leftarrow (1/|\mathcal{U}|) \sum_{i \in \mathcal{U}} \ell_i$
  **return** $\mathbf{U}, \mathbf{V}, \xi, \{\ell_i\}_{i \in \mathcal{U}}, \mathbf{G}_V$

**Function** $\texttt{ComputeXi}(\xi, \{\ell_i\}_{i \in \mathcal{U}})$:
  **for** $l \leftarrow 1$ **to** $L$ **do**
    Uniformly draw $\mathcal{U}_b \subseteq \mathcal{U}$
    $\widehat{d} \leftarrow \frac{\nabla_\xi \widehat{\Psi}_{1-\alpha}(\xi)}{\nabla_\xi^2 \widehat{\Psi}_{1-\alpha}(\xi)}$
    Backtracking line search of $\gamma$,
    $\xi \leftarrow \xi - \gamma \cdot \widehat{d}$
  **return** $\xi$

## 4 RELATED WORK

**Risk-averse optimization.** Coherent risk measures such as CVaR (also called expected short-fall (Acerbi and Tasche, 2002)) are widely used instead of (incoherent) VaR (Artzner, 1997; Artzner et al., 1999; Jorion, 2007). Despite the desirable properties of CVaR (Pflug, 2000), its optimization is often found challenging owing to the non-smooth check function. Thus, in CVaR optimization and statistical learning with CVaR (Curi et al., 2020; Mhammedi et al., 2020; Soma and Yoshida, 2020), smoothing techniques are utilized, e.g., piece-wise quadratic smoothed plus (Alexander et al., 2006; Xu and Zhang, 2009) and softplus (Soma and Yoshida, 2020). Distributionally robust optimization (DRO) is another prevalent approach to risk-averse learning (Rahimian and Mehrotra, 2019). Recent studies explore dual-free DRO algorithms to avoid maintaining large dual variables (Jin et al., 2021; Levy et al., 2020; Qi et al., 2021). The dual-free algorithms introduce a Bregman distance defined on dual variables (Lan and Zhou, 2018; Wang and Xiao, 2017). By contrast, we apply convolution-type smoothing to CVaR for the efficient computation of $\xi$ and the separability via primal-dual splitting.

**Quantile regression.** In contrast to the least squares regression that models a conditional mean, quantile regression (QR) offers more flexibility to model the entire conditional distribution (Koenker and Bassett Jr, 1978; Koenker and Hallock, 2001; Koenker et al., 2017). Our study is related to the computational aspects of QR that involve the piece-wise linear check function. Gu et al. (2018) proposed a method based on the alternating direction method of multipliers (ADMM) (Boyd et al., 2011) for smoothness. However, this approach is not suited for our case because of non-separability for items in the penalty term of the augmented Lagrangian (See Appendix F). In line with Horowitz's smoothing (Horowitz, 1998), recent studies developed methods using convolution-type smoothing for large-scale inference (Fernandes et al., 2021). These studies realize tractable estimation using ADMM (Tan et al., 2022), Frisch-Newton algorithm (He et al., 2021), and proximal gradient method (Man et al., 2022), whereas these QR methods cannot be used for our setting.

**Robust recommender systems.** To robustify recommender systems against the behavior of malicious minority users who have an incentive to bias data for economic reasons, previous studies (Mehta and Nejdl, 2008; Mehta et al., 2007) proposed methods using MF and Huber's M-estimator (Huber, 2011), which is closely related to QR. There is a growing interest in controlling the performance distribution of a recommender model, such as fairness-aware recommendation, where fairness towards users is essential (Do et al., 2021; Patro et al., 2020). Several recent studies have explored improving semi-worst-case performance for users. Singh et al. (2020) introduced a multi-objective optimization approach that balances reward and CVaR-based healthiness for online recommendation. Wen et al. (2022) proposed a method based on group DRO over given user groups instead of individual users. By contrast, to optimize the worst-case performance for individual users, Shivaswamy and Garcia-Garcia (2022) examined an adversarial learning approach, which trains two models. However, these methods do not focus on practical scalability and rely on gradient descent to directly optimize their objectives.

## 5 NUMERICAL EXPERIMENTS

### 5.1 EXPERIMENTAL SETUP

**Datasets and evaluation protocol.** We experiment with two MovieLens datasets (***ML-1M*** and ***ML-20M***) (Harper and Konstan, 2015) and Million Song Dataset (***MSD***) (Bertin-Mahieux et al., 2011). We strictly follow the standard evaluation methodology (Liang et al., 2018; Rendle et al., 2022; Steck, 2019b; Weimer et al., 2007) based on *strong generalization*. To generate implicit feedback datasets. we retain interactions with ratings larger than 4 of MovieLens datasets, while we use all interactions for ***MSD***. We then consider 80% of users for training (i.e., $\{\mathcal{V}_i\}_{i \in \mathcal{U}}$). The remaining 10% of users in two holdout splits are used for validation and testing. During the evaluation phases, the 80% interactions of each user are disclosed to a model as input to make predictions for the user, and the remaining 20% are used to compute ranking measures. We use Recall@$K$ (R@$K$) as the quality measure of a ranked list and take the average over all testing users. We also evaluate the performance for semi-worst-case scenarios by considering the mean R@$K$ for worse-off users whose R@$K$ is lower than the $\alpha$-quantile among the testing users; note that setting $\alpha = 1.0$ is equivalent to the average case. The mean R@$K$ with $\alpha = 1.0$ may be referred to as R@$K$ for simplicity. For the validation measure, we used R@20 for ***ML-1M*** and R@50 for ***ML-20M*** and ***MSD***.

**Models.** We compare SAFER$_2$ to iALS (Rendle et al., 2022), ERM-MF in Eq. (3), and CVaR-MF in Eq. (5). To set strong baselines, we consider Mult-VAE (Liang et al., 2018) and a recent iALS variant with Tikhonov regularization, which is competitive to the state-of-the-art methods (Rendle et al., 2022). We do not compare pairwise ranking methods (e.g., (Rendle et al., 2009; Weston et al., 2011)) as they are known to be non-competitive on the three datasets (Liang et al., 2018; Rendle et al., 2022; Sedhain et al., 2016). We consider the instance of SAFER$_2$ with a Gaussian kernel $k_h(u) = (2\pi h^2)^{-1/2} \exp(-u^2/2h^2)$. In all models, we initialize $\mathbf{U}$ and $\mathbf{V}$ with Gaussian noise with standard deviation $\sigma/\sqrt{d}$ where $\sigma = 0.1$ in all datasets (Rendle et al., 2022) and tune $\beta_0$ and $\lambda$. We set $\alpha = 0.3$ in Eq. (5) and Eq. (7). For SAFER$_2$, we also search the bandwidth $h$ and set the number of NR iterations as $L = 5$. For CVaR-MF, we tune a global learning rate for all variables in the batch subgradient method. For Mult-VAE, we tune the learning rate, batch size, and annealing parameter. For ***ML-20M*** and ***MSD***, we use the sub-sampled NR algorithm in the $\xi$ step with sampling ratio $|\mathcal{U}_b|/|\mathcal{U}| = 0.1$ for SAFER$_2$. The dimensionality $d$ of user/item embeddings is set to 32, 256, and 512 for ***ML-1M***, ***ML-20M***, and ***MSD***, respectively; for Mult-VAE, we use a three-layer architecture $[|\mathcal{V}| \to d \to d - 1 \to |\mathcal{V}|]$ because the encoded representation can be viewed as $d$-dimensional user embeddings with an auxiliary dimension of one for the bias term in the last fully-connected layer. During the validation and testing phases, each method optimizes an embedding (i.e., $\mathbf{u}$) of each user based on the user's 80% interactions while fixing trained $\mathbf{V}$, and predicts the scores for all items. Notably, for each testing user, CVaR-MF and SAFER$_2$ solve the objective of ERM-MF in Eq. (3) as each user's subproblem is separable and can be solved independently. Our source code is publicly available at `https://github.com/riktor/safer2-recommender`. Further detailed descriptions and additional results are provided in Appendix G.

### 5.2 RESULTS

**Benchmark evaluation.** The evaluation results for the three datasets are summarized in Table 1. For the semi-worst-case scenarios ($\alpha = 0.3$), SAFER$_2$ shows excellent quality in most cases, whereas baselines exhibit a decline in some cases. The quality of CVaR-MF deteriorates in all settings, emphasizing the advantage of our smoothing approach. The average-case performance ($\alpha = 1.0$) of SAFER$_2$ is also remarkable, which may be due to its robustness and generalization ability for limited testing samples. In fact, for the largest ***MSD*** with 50,000 testing users, SAFER$_2$ performs slightly worse than iALS in R@50 with $\alpha = 1.0$. To break down the above results of ***ML-20M*** and ***MSD***, Figure 3 compares the relative performance of each method with respect to iALS at each quantile level $\alpha$. Here, because the instances of R@$K$ with $\alpha$ vary on different scales depending on $K$ and $\alpha$, we show the relative performance of each method over iALS, obtained by dividing the method's performance by that of iALS, in the y-axis of each figure. We can see the clear improvement of SAFER$_2$ for smaller $\alpha$ while it maintains the average performance. It is particularly remarkable that SAFER$_2$ outperforms Mult-VAE in terms of R@50 for tail users in ***ML-20M***.

Table 1: Ranking quality comparison on *ML-1M*, *ML-20M*, and *MSD*.

| | *ML-1M* | | | | *ML-20M* | | | | *MSD* | | | |
|---|---|---|---|---|---|---|---|---|---|---|---|---|
| | R@20 | R@50 | R@20 | R@50 | R@50 | R@100 | R@50 | R@100 | R@50 | R@100 | R@50 | R@100 |
| **Models** | $\alpha=1.0$ | $\alpha=1.0$ | $\alpha=0.3$ | $\alpha=0.3$ | $\alpha=1.0$ | $\alpha=1.0$ | $\alpha=0.3$ | $\alpha=0.3$ | $\alpha=1.0$ | $\alpha=1.0$ | $\alpha=0.3$ | $\alpha=0.3$ |
| iALS | 0.3450 | 0.4697 | 0.1166 | 0.2391 | 0.5263 | 0.6448 | 0.2085 | 0.3264 | **0.3590** | 0.4604 | 0.0963 | 0.1684 |
| Mult-VAE | 0.3329 | 0.4539 | 0.1106 | 0.2245 | **0.5313** | **0.6565** | 0.2091 | **0.3376** | 0.3482 | 0.4347 | 0.0854 | 0.1485 |
| ERM-MF | 0.3448 | 0.4700 | 0.1189 | 0.2315 | 0.5275 | 0.6441 | 0.2088 | 0.3251 | 0.3544 | 0.4540 | 0.0945 | 0.1644 |
| CVaR-MF | 0.3318 | 0.4495 | 0.1061 | 0.2257 | 0.5031 | 0.6277 | 0.1975 | 0.3187 | 0.3234 | 0.4121 | 0.0781 | 0.1412 |
| SAFER$_2$ | **0.3517** | **0.4804** | **0.1279** | **0.2428** | 0.5308 | 0.6501 | 0.2152 | 0.3342 | 0.3585 | **0.4605** | **0.0983** | **0.1700** |
| **Dataset** | 6,168 users | | 2,811 movies | | 136,677 users | | 20,108 movies | | 571,355 users | | 41,140 songs | |
| **statistics** | 0.56 M interactions | | | | 9.54 M interactions | | | | 33.6 M interactions | | | |

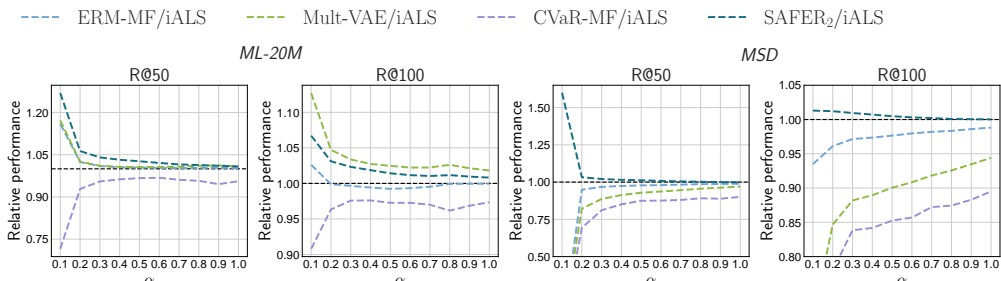

Figure 3: Relative performance of each method over iALS for each quantile level.

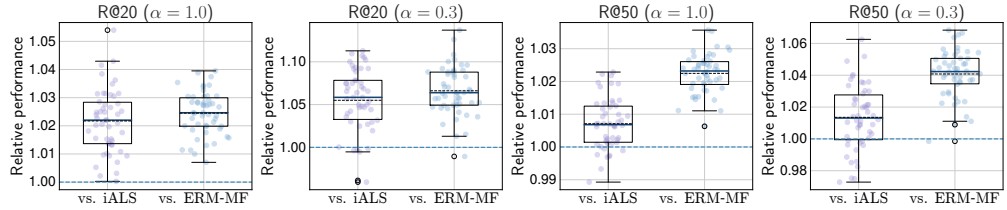

Figure 4: Distributions of relative performances of SAFER$_2$ over ERM-MF and iALS on *ML-1M*.

**Robustness.** Considering the unexpected improvement of SAFER$_2$ for the average scenario on *ML-1M*, we conduct further analysis of the average performance for limited testing users. We repeatedly evaluate each model using the aforementioned protocol on 50 data splits independently generated from *ML-1M* with different random seeds. This evaluation procedure can be considered as a nested cross-validation (Cawley and Talbot, 2010) with 50 outer folds and 2 inner folds. The resulting relative performances of SAFER$_2$ over ERM-MF and iALS[5] are shown in Figure 4, with each point indicating the ratio of testing measurements for a particular data split. Each measurement is obtained by taking an average of 10 models with different initialization weights on the same split. SAFER$_2$ generally exhibits a superior quality, even for the average cases. Furthermore, its advantage in the case with $\alpha = 0.3$ highlights the robustness of the proposed approach.

**Runtime comparison.** The runtime per epoch of each method[6] is shown in Table 2. All experiments of MF-based methods (i.e., iALS, ERM-MF, CVaR-MF, and SAFER$_2$) are performed using our multi-threaded C++ implementation, originally provided by Rendle et al. (2022), which utilizes Eigen to perform vector/matrix operations that support AVX instructions. The reported numbers are the averaged runtime through

Table 2: Runtime per epoch.

| | *ML-20M* | *MSD* |
|---|---|---|
| **Models** | Runtime/epoch | Runtime/epoch |
| iALS | 3.16 sec | 53.5 sec |
| Mult-VAE | 9.06 sec | 112.5 sec |
| CVaR-MF | 1.67 sec | 25.2 sec |
| SAFER$_2$ | 3.45 sec | 57.0 sec |

50 epochs measured using 86.4 GB RAM and Intel(R) Xeon(R) CPU @ 2.00GHz with 96 CPU cores. We implemented Mult-VAE using PyTorch and utilized an NVIDIA P100 GPU to speed up its training. We here consider a variant of CVaR-MF with the Adam preconditioner (Kingma and Ba, 2014) as a baseline. The runtime of SAFER$_2$ is competitive to iALS, which is the most efficient

---

[5]We omitted CVaR-MF and Mult-VAE as this protocol is costly due to their slow convergence.

[6]We omitted ERM-MF as it is nearly identical to iALS.

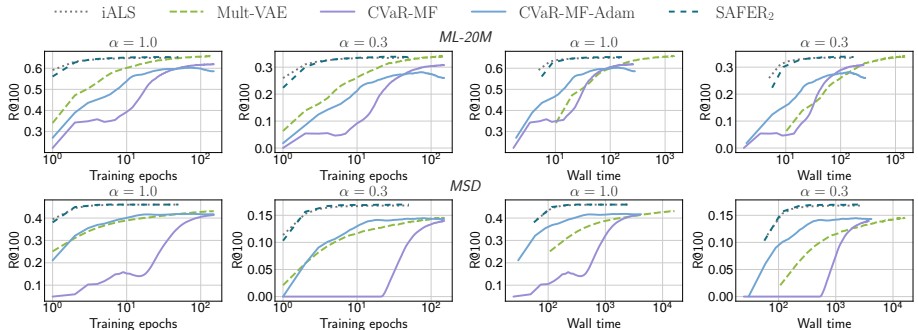

Figure 5: Ranking quality vs. training epochs/wall time on **ML-20M** (top) and **MSD** (bottom).

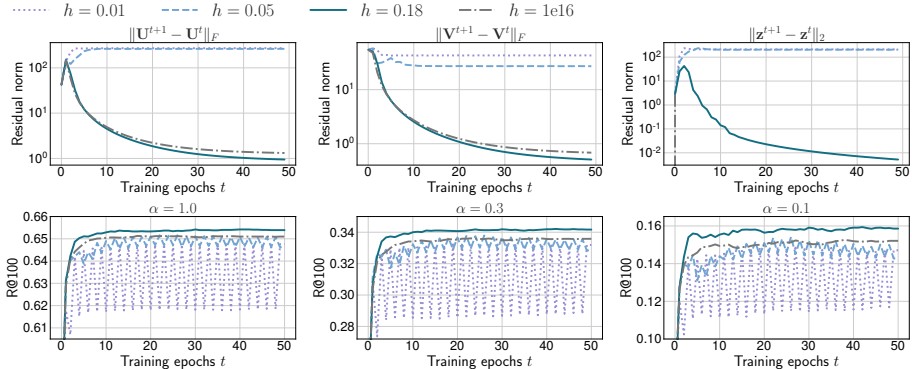

Figure 6: Convergence profile of SAFER$_2$ with different bandwidth $h$ on **ML-20M**.

CF method. CVaR-MF is faster than SAFER$_2$ and iALS in terms of runtime per epoch because it is free from the cubic dependency on $d$. However, as shown in Figure 5, CVaR-MF requires much more training epochs to obtain acceptable performance even with the Adam preconditioner; it has not converged yet even with 50 epochs while iALS and SAFER$_2$ converge with around 10 epochs. Mult-VAE is inefficient in terms of both runtime/epoch and training epochs even with GPU; it thus shows very slow convergence in terms of wall time (the right side of Figure 5). These results show that SAFER$_2$ enables efficient optimization in terms of both runtime and convergence speed.

**Convergence profile.** Lacking a theoretical convergence guarantee, we empirically investigate the training behavior of SAFER$_2$. Figure 6 demonstrates the effect of bandwidth $h$ on the convergence profile of SAFER$_2$. The top row of the figures shows the residual norms of each block at each step, while the bottom row illustrates the validation measures. We observe that setting a small bandwidth ($h = 0.01, 0.05$) impedes convergence; in particular, the case with $h = 0.01$, which almost degenerates to the non-smooth CVaR, experiences fluctuation in residual norms and ranking quality. By contrast, a sufficiently large bandwidth ($h = 0.18, 1e16$) ensures stable convergence, whereas $h = 1e16$, which is reduced to ERM, does not achieve optimal performance in semi-worst-case scenarios. These results support that convolution-type smoothing is vital for SAFER$_2$.

## 6 CONCLUSION

Towards the modernization of industrial recommender systems, where the engagement of tail users is vital for business growth, we presented a practical algorithm that ensures high-quality personalization for each individual user while maintaining scalability for real-world applications. Our algorithm called SAFER$_2$ overcomes non-smoothness and non-separability in CVaR minimization by using convolution-type smoothing which is the essential ingredient to obtain its separable reformulation and attain scalability over two dimensions of users and items. Compared to the celebrated iALS, our SAFER$_2$ is scalable to the same extent yet exhibits superior semi-worst-case performance without sacrificing average quality.

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

# Appendix

## Table of Contents

## A  CONVEX AND SEPARABLE UPPER BOUND OF PAIRWISE LOSS

We first derive the differentiable upper bound of the loss function in Eq. (1).

$$\frac{1}{|\mathcal{V}_i|} \sum_{j \in \mathcal{V}_i} \sum_{j' \in \mathcal{V}} \mathbb{1}\{f_\theta(i, j') \geq f_\theta(i, j)\},$$

$$\leq \frac{1}{|\mathcal{V}_i|} \sum_{j \in \mathcal{V}_i} \sum_{j' \in \mathcal{V}} \max(0, f_\theta(i, j') - f_\theta(i, j) + 1)$$

$$\leq \frac{1}{|\mathcal{V}_i|} \sum_{j \in \mathcal{V}_i} \sum_{j' \in \mathcal{V}} \left[f_\theta(i, j') - f_\theta(i, j) + 1\right]^2.$$

Note that we used $(x + 1)^2 \geq \mathbb{1}\{x \geq 0\}$ to obtain the third inequality.

We next derive a separable upper bound of the above loss as follows:

$$\frac{1}{|\mathcal{V}_i|} \sum_{j \in \mathcal{V}_i} \sum_{j' \in \mathcal{V}} \left[ f_\theta(i, j') - f_\theta(i, j) + 1 \right]^2$$

$$\leq \frac{1}{|\mathcal{V}_i|} \sum_{j \in \mathcal{V}_i} \sum_{j' \in \mathcal{V}} \left( 1 - [f_\theta(i, j)]^2 + f_\theta(i, j')^2 \right)$$

$$= \frac{1}{|\mathcal{V}_i|} \sum_{j \in \mathcal{V}_i} \left( |\mathcal{V}| \cdot [1 - f_\theta(i, j)]^2 + \sum_{j \in \mathcal{V}} f_\theta(i, j)^2 \right)$$

$$\leq |\mathcal{V}| \cdot \left( \frac{1}{|\mathcal{V}_i|} \sum_{j \in \mathcal{V}_i} [1 - f_\theta(i, j)]^2 + \frac{\mu}{|\mathcal{V}|} \cdot \sum_{j \in \mathcal{V}} f_\theta(i, j)^2 \right), \tag{16}$$

where $\mu \geq 1$ is a hyperparameter. By substituting $\beta_0 = \mu/|\mathcal{V}| \geq 1/|\mathcal{V}|$, we obtain our proposed loss.[7] The obtained loss is convex w.r.t. the predicted score $f_\theta(i, j)$ and separable w.r.t. $f_\theta(i, j)$ and $f_\theta(i, j')$ if $j \neq j'$. It also implies that, when using an MF model, $f_\theta(i, j) = \langle \mathbf{u}_i, \mathbf{v}_j \rangle$, the loss is separable and block multi-convex w.r.t. $\mathbf{u}_i$, $\mathbf{v}_j$ and $\mathbf{v}_{j'}$ for all $(j, j') \in \mathcal{V}^2$ such that $j \neq j'$.

## B  IMPLICIT ALTERNATING LEAST SQUARES (iALS)

In this appendix, we provide a brief description of iALS (Hu et al., 2008; Rendle et al., 2022). The objective of iALS can be considered as a variant of ERM-MF, which is defined as follows:

$$\min_{\mathbf{U}, \mathbf{V}} \sum_{i \in \mathcal{U}} \ell_{\text{iALS}}(\mathbf{V} \mathbf{u}_i, \mathcal{V}_i) + \Omega(\mathbf{U}, \mathbf{V}), \tag{17}$$

where

$$\ell_{\text{iALS}}(\mathbf{V} \mathbf{u}_i, \mathcal{V}_i) \coloneqq \sum_{j \in \mathcal{V}_i} \frac{1}{2} (\mathbf{u}_i^\top \mathbf{v}_j - 1)^2 + \frac{\beta_0}{2} \|\mathbf{V} \mathbf{u}_i\|_2^2. \tag{18}$$

Observe that iALS's loss function is unnormalized for each user in contrast to the loss in Eq. (4). Using this, iALS solves the above optimization problem by alternatingly solving the block-wise subproblems as follows:

$$\mathbf{U}^{(t+1)} = \underset{\mathbf{U}}{\arg\min} \left\{ \sum_{i \in \mathcal{U}} \ell_{\text{iALS}}(\mathbf{V}^{(t)} \mathbf{u}_i, \mathcal{V}_i) + \frac{1}{2} \|\mathbf{\Lambda}_u^{1/2} \mathbf{U}\|_F^2 \right\}, \tag{19}$$

$$\mathbf{V}^{(t+1)} = \underset{\mathbf{V}}{\arg\min} \left\{ \sum_{i \in \mathcal{U}} \ell_{\text{iALS}}(\mathbf{V} \mathbf{u}_i^{(t+1)}, \mathcal{V}_i) + \frac{1}{2} \|\mathbf{\Lambda}_v^{1/2} \mathbf{V}\|_F^2 \right\}. \tag{20}$$

Each subproblem is row-wise separable, and we can obtain a closed-form update for each row of $\mathbf{U}$ and $\mathbf{V}$ by solving the following linear systems,

$$\left( \sum_{j \in \mathcal{V}_i} \mathbf{v}_j^{(t)} \otimes \mathbf{v}_j^{(t)} + \beta_0 \sum_{j \in \mathcal{V}} \mathbf{v}_j^{(t)} \otimes \mathbf{v}_j^{(t)} + \lambda_u^{(i)} \mathbf{I} \right) \mathbf{u}_i = \sum_{j \in \mathcal{V}_i} \mathbf{v}_j^{(t)}, \tag{21}$$

$$\left( \sum_{i \in \mathcal{U}_j} \mathbf{u}_i^{(t+1)} \otimes \mathbf{u}_i^{(t+1)} + \beta_0 \sum_{i \in \mathcal{U}} \mathbf{u}_i^{(t+1)} \otimes \mathbf{u}_i^{(t+1)} + \lambda_v^{(j)} \mathbf{I} \right) \mathbf{v}_j = \sum_{i \in \mathcal{U}_j} \mathbf{u}_i^{(t+1)}, \tag{22}$$

where $\mathbf{x} \otimes \mathbf{x} = \mathbf{x}\mathbf{x}^\top$ is the outer product operator. Various regularization strategies have been proposed for iALS (Rendle et al., 2022; Zhou et al., 2008). Rendle et al. (2022) proposed the following weighting strategy,

$$(\mathbf{\Lambda}_u)_{i,i} = \lambda_u^{(i)} = \lambda \left( |\mathcal{V}_i| + \beta_0 |\mathcal{V}| \right)^\nu, \quad (\mathbf{\Lambda}_v)_{j,j} = \lambda_v^{(j)} = \lambda \left( |\mathcal{U}_j| + \beta_0 |\mathcal{U}| \right)^\nu. \tag{23}$$

In this paper, we consider $\nu = 1$ as recently suggested by Rendle et al. (2022).

---

[7]The second term on the RHS of the final inequality is known as the implicit regularizer (Bayer et al., 2017) and the gravity term (Krichene et al., 2018).

## C  CONVOLUTION-TYPE SMOOTHING

### C.1  DEFINITION AND DERIVATIVES

We define the ramp function $\rho_1(\cdot)$ and the check function $\rho_\tau(\cdot)$ for $\tau \in (0,1)$ as follows:

$$\rho_1(u) = \begin{cases} 0, & u \leq 0 \\ u, & u > 0 \end{cases} \quad \text{and} \quad \rho_\tau(u) = \begin{cases} (\tau - 1)u, & u \leq 0 \\ \tau u, & u > 0 \end{cases}. \tag{24}$$

Given $\alpha \in (0,1)$, we consider the convolution function $(\rho_{1-\alpha} * k_h)$ with the kernel density function $k_h(\cdot) := h^{-1}k(\cdot/h)$ with bandwidth $h > 0$. We can show that

$$
\begin{aligned}
(\rho_{1-\alpha} * k_h)(u) &= \int \rho_{1-\alpha}(v)k_h(v-u)dv \\
&= -\alpha \int_{-\infty}^{0} v \cdot k_h(v-u)dv + (1-\alpha)\int_{0}^{\infty} v \cdot k_h(v-u)dv \\
&= -\alpha \int_{-\infty}^{\infty} v \cdot k_h(v-u)dv + \int_{0}^{\infty} v \cdot k_h(v-u)dv \\
&= -\alpha \int_{-\infty}^{\infty} (t+u) \cdot k_h(t)dt + \int_{-u}^{\infty} (t+u) \cdot k_h(t)dt,
\end{aligned}
$$

where an application of a change of variables yields the last equality. Then, we can obtain the first derivative of the convolution function as follows:

$$
\begin{aligned}
\nabla_u (\rho_{1-\alpha} * k_h)(u) &= -\alpha \int_{-\infty}^{\infty} k_h(t)dt + \int_{-u}^{\infty} k_h(t)dt \\
&= -\alpha + \left(1 - \int_{-\infty}^{-u} k_h(t)dt\right) \\
&= (1-\alpha) - K_h(-u). 
\end{aligned} \tag{25}
$$

Also, we can obtain the second derivative

$$\nabla_u^2 (\rho_{1-\alpha} * k_h)(u) = k_h(-u). \tag{26}$$

### C.2  INTERPRETATION

In this subsection, we offer a general understanding of convolution smoothing as a kernel density estimation for the underlying errors. Let $y$ be a continuous random variable having the density function $f_y$. We consider the problem of predicting $y$ by a parametric model $g_\theta(x)$ with parameters $\theta$ and predictors $x$. Given some function $\rho : \mathbb{R} \to [0, \infty)$, we consider a population minimization problem,

$$\min_\theta \mathbb{E}[\rho(y - g_\theta(x))]. \tag{27}$$

Applying the change of variables, we can show that

$$
\begin{aligned}
\mathbb{E}[\rho(y - g_\theta(x))] &= \int \rho(y - g_\theta(x))f_y(y)dy \\
&= \int \rho(v)f_u(v;\theta)dv,
\end{aligned} \tag{28}
$$

where $f_u(v;\theta) := f_y(v + g_\theta(x))$ can be considered as a density of $u(\theta) := y - g_\theta(x)$.

Given finite samples $\{(x_i, y_i)\}_{i=1}^n$ with the sample size being $n$, we can non-parametrically estimate the density $f_u(v;\theta)$ by the kernel estimator, defined by

$$\widehat{f}_u(v;\theta) := \frac{1}{n}\sum_{i=1}^{n} k_h(v - u_i(\theta)), \tag{29}$$

with a kernel density function $k_h(\cdot) := h^{-1}k(\cdot/h)$ with bandwidth $h > 0$. Then, we can write the finite-sample counterpart of the objective function as

$$\int \rho(v)\widehat{f}_u(v;\theta)dv = \frac{1}{n}\sum_{i=1}^{n}\int \rho(v)k_h(v - u_i(\theta))dv$$

$$= \frac{1}{n}\sum_{i=1}^{n}(\rho * k_h)(u_i(\theta)), \tag{30}$$

where the last equality is due to the definition of convolution-type smoothing. The result above show that the convolution smoothing can be interpreted as a kernel density estimation technique for approximating the distribution of underlying errors.

### C.3 ON THE CONVOLUTION-TYPE SMOOTHING FOR CVAR

**Proposition C.1.** *If the kernel function $k_h$ is symmetric, the subproblem of $\xi$ is equivalent to the following,*

$$\min_{\xi}\left\{\sum_{i\in\mathcal{U}}[(\rho_{1-\alpha} * k_h)(\ell(\mathbf{V}\mathbf{u}_i, \mathcal{V}_i) - \xi)]\right\},$$

*where $\rho_{1-\alpha}(u) = (1 - \alpha - \mathbb{1}\{u < 0\})u$.*

*Proof.* We show that, if the kernel function $k_h$ is symmetric, the following equality holds,

$$\xi + \frac{1}{\alpha}(\rho_1 * k_h)(y - \xi) = y + \frac{1}{\alpha}(\rho_{1-\alpha} * k_h)(y - \xi). \tag{31}$$

From the definition of the convolution operator, we have

$$(\rho_1 * k_h)(u) = \int \max(0, v) \cdot k_h(v - u)dv = \int_0^\infty v \cdot k_h(v - u)dv, \tag{32}$$

which implies

$$\xi + \frac{1}{\alpha}(\rho_1 * k_h)(y - \xi) = \xi + \frac{1}{\alpha}\int_0^\infty v \cdot k_h(v - \{y - \xi\})dv. \tag{33}$$

Here, we also have

$$(\rho_{1-\alpha} * k_h)(u) = -\alpha\int_{-\infty}^\infty v \cdot k_h(v - u)dv + \int_0^\infty v \cdot k_h(v - u)dv. \tag{34}$$

It follows that

$$y - \xi + \frac{1}{\alpha}(\rho_{1-\alpha} * k_h)(y - \xi)$$

$$= y - \xi - \int_{-\infty}^\infty v \cdot k_h(v - \{y - \xi\})dv + \frac{1}{\alpha}\int_0^\infty v \cdot k_h(v - \{y - \xi\})dv$$

$$= \int_{-\infty}^\infty (u - v) \cdot k_h(v - \{y - \xi\})dv + \frac{1}{\alpha}\int_0^\infty v \cdot k_h(v - \{y - \xi\})dv$$

$$= \frac{1}{\alpha}\int_0^\infty v \cdot k_h(v - \{y - \xi\})dv, \tag{35}$$

where the second equality is due to that $\int_{-\infty}^\infty k_h(v - \{y - \xi\})dv = 1$ and the third equality is due to the symmetric kernel. By combining Eq. (33) and Eq. (35), we have

$$\xi + \frac{1}{\alpha}(\rho_1 * k_h)(y - \xi) = \xi + y - \xi + \frac{1}{\alpha}(\rho_{1-\alpha} * k_h)(y - \xi) \tag{36}$$

$$\Longleftrightarrow \xi + \frac{1}{\alpha}(\rho_1 * k_h)(y - \xi) = y + \frac{1}{\alpha}(\rho_{1-\alpha} * k_h)(y - \xi), \tag{37}$$

which completes the proof. $\qquad\square$

**ERM-QE Decomposition.** The above observation immediately implies Proposition C.1,

$$\underbrace{\xi + \frac{1}{\alpha|\mathcal{U}|} \sum_{i \in \mathcal{U}} (\rho_1 * k_h)(\ell(\mathbf{V}\mathbf{u}_i, \mathcal{V}_i) - \xi)}_{\Psi_{1-\alpha}(\mathbf{U}, \mathbf{V}, \xi)} = \underbrace{\frac{1}{|\mathcal{U}|} \sum_{i \in \mathcal{U}} \ell(\mathbf{V}\mathbf{u}_i, \mathcal{V}_i)}_{\text{ERM}} + \underbrace{\frac{1}{\alpha|\mathcal{U}|} \sum_{i \in \mathcal{U}} (\rho_{1-\alpha} * k_h)(\ell(\mathbf{V}\mathbf{u}_i, \mathcal{V}_i) - \xi)}_{\text{Smoothed quantile estimation}}.$$

(38)

This result is also interesting in the sense that the CVaR objective $\Psi_{1-\alpha}(\mathbf{U}, \mathbf{V}, \xi)$ can be decomposed into the objectives of ERM and smoothed quantile estimation. However, we do not use the RHS for the update of $\mathbf{U}$ and $\mathbf{V}$ because the term of smoothed quantile estimation is not block convex because $\rho_{1-\alpha}(u)$ is convex but decreasing in $u < 0$; See also Figure 2.

### C.4 Properties of Convolution-Type Smoothed Check Functions

We here discuss the properties of smoothed check functions assumed to derive SAFER$_2$.

**Property C.2.** *For any bandwidth $h > 0$ and any quantile level $\alpha \in [0, 1]$, the smoothed check function $(\rho_1 * k_h)$ satisfies the following properties:*

*(1) $(\rho_1 * k_h)$ is non-decreasing.*

*(2) $(\rho_1 * k_h)$ is closed if the loss function $\ell$ and smoothed quantile $\xi$ take finite values.*

*(3) $(\rho_1 * k_h)$ is strictly convex if the kernel function $k_h$ has the full support.*

*Proof.* The first derivative of the smoothed check function is $\nabla_u(\rho_1 * k_h)(u) = 1 - K_h(-u)$, and therefore, it is non-negative based on the definition of the CDF $K_h$. This implies (1). By satisfying the assumptions where (a) the function $(\rho_{1-\alpha} * k_h)$ is continuous and (b) its domain $\text{dom}(\rho_{1-\alpha} * k_h)$ is closed, (2) holds. Moreover, since the second derivative of the smoothed check function is the density function (i.e., $\nabla_u^2(\rho_{1-\alpha} * k_h)(u) = k_h(u)$), if $k_h(u) > 0$ holds for the domain of $u$, then (3) immediately follows. $\square$

In (1), we confirm that the smoothed check function maintains the non-decreasing property. Consequently, the composite function $(\rho_1 * k_h) \circ \ell$ preserves block convexity if $\ell$ is block convex. By using (1)-(2), the biconjugate of $(\rho_1 * k_h)$ is $(\rho_1 * k_h)$ itself (Boyd and Vandenberghe, 2004), enabling the separable reformulation in Eq. (11); note that the assumptions of finite $\ell$ and $\xi$ for (2) may not be stringent conditions in practice. (3) implies that the Newton–Raphson method in the $\xi$ step may require full-support kernels, such as logistic and Gaussian kernels. However, we can still use the kernels with the support on $(-1, 1)$, such as uniform and Epanechnikov kernels, by using a sufficiently large bandwidth, which allows us to expand the support on $(-h, h)$; we further discuss the instantiation of SAFER$_2$ with such kernel functions in Appendix D.3.

---

**Algorithm 3:** SAFER$_2$ solver.

---

$\forall i \in \mathcal{U}, \ \mathbf{u}_i \sim \mathcal{N}(\vec{0}, (\sigma/\sqrt{d})\mathbf{I}_d), \quad \forall j \in \mathcal{V}, \ \mathbf{v}_j \sim \mathcal{N}(\vec{0}, (\sigma/\sqrt{d})\mathbf{I}_d)$

$\xi \leftarrow 0$

**for** $t \leftarrow 1$ **to** $T$ **do**

   $\mathbf{G}_V \leftarrow \mathbf{V}^\top\mathbf{V} = \sum_{j\in\mathcal{V}} \mathbf{v}_j\mathbf{v}_j^\top$

   Compute $\forall i \in \mathcal{U}, \ \ell_i = \frac{1}{2|\mathcal{V}_i|}\sum_{j\in\mathcal{V}_i}(1-\mathbf{u}_i^\top\mathbf{v}_j)^2 + \frac{\beta_0}{2}\cdot\mathbf{u}_i^\top\mathbf{G}_V\mathbf{u}_i$

   **for** $l \leftarrow 1$ **to** $L$ **do**

      Uniformly draw $\mathcal{U}_b \subseteq \mathcal{U}$

      $\widehat{d} \leftarrow \dfrac{\nabla_\xi\widehat{\Psi}_{1-\alpha}(\mathbf{U},\mathbf{V},\xi)}{\nabla_\xi^2\widehat{\Psi}_{1-\alpha}(\mathbf{U},\mathbf{V},\xi)}$

      $\gamma \leftarrow \arg\max_{\gamma\in(0,1)}\gamma$

         s.t. $\widehat{\Psi}_{1-\alpha}(\mathbf{U},\mathbf{V},\xi-\gamma\widehat{d}) \leq \widehat{\Psi}_{1-\alpha}(\mathbf{U},\mathbf{V},\xi) - c\gamma\widehat{d}\nabla_\xi\widehat{\Psi}_{1-\alpha}(\mathbf{U},\mathbf{V},\xi-\gamma\widehat{d})$

      $\xi \leftarrow \xi - \gamma\cdot\widehat{d}$

   **end**

   **for** $i \leftarrow 1$ **to** $|\mathcal{U}|$ **do**

      $z_i \leftarrow 1 - K_h(\xi - \ell_i)$

      $\mathbf{u}_i \leftarrow \left(\frac{z_i}{|\mathcal{V}_i|}\sum_{j\in\mathcal{V}_i}\mathbf{v}_j\mathbf{v}_j^\top + z_i\beta_0\mathbf{G}_V + \alpha|\mathcal{U}|\lambda_u^{(i)}\mathbf{I}_d\right)^{-1}\frac{z_i}{|\mathcal{V}_i|}\sum_{j\in\mathcal{V}_i}\mathbf{v}_j$

   **end**

   $\widetilde{\mathbf{G}}_U \leftarrow \mathbf{U}^\top\text{diagMat}(\mathbf{z})\mathbf{U}$

   **for** $j \leftarrow 1$ **to** $|\mathcal{V}|$ **do**

      $\mathbf{v}_j \leftarrow \left(\sum_{i\in\mathcal{U}_j}\frac{z_i}{|\mathcal{V}_i|}\mathbf{u}_i\mathbf{u}_i^\top + \beta_0\widetilde{\mathbf{G}}_U + \alpha|\mathcal{U}|\lambda_v^{(j)}\mathbf{I}_d\right)^{-1}\sum_{i\in\mathcal{U}_j}\frac{z_i}{|\mathcal{V}_i|}\mathbf{u}_i$

   **end**

**end**

---

## D   Implementation details of SAFER$_2$

This appendix provides a detailed description of SAFER$_2$, including its various instances for kernel functions. Furthermore, we shall detail a variant of SAFER$_2$ for a large embedding size, utilizing the subspace-based block coordinate descent as recently introduced by Rendle et al. (2021).

**Alternating optimization.** SAFER$_2$ updates each block cyclically as follows:

$$\xi^{(t+1)} = \arg\min_{\xi}\left\{\xi + \frac{1}{\alpha|\mathcal{U}|}\sum_{i\in\mathcal{U}}(\rho_1 * k_h)(\ell(\mathbf{V}^{(t)}\mathbf{u}_i^{(t)}, \mathcal{V}_i) - \xi)\right\}, \tag{39}$$

$$\mathbf{z}^{(t+1)} = \arg\max_{\mathbf{z}}\left\{\sum_{i\in\mathcal{U}}\left[z_i\cdot(\ell(\mathbf{V}^{(t)}\mathbf{u}_i^{(t)}, \mathcal{V}_i) - \xi^{(t+1)}) - (\rho_1 * k_h)^*(z_i)\right]\right\}, \tag{40}$$

$$\mathbf{U}^{(t+1)} = \arg\min_{\mathbf{U}}\left\{\frac{1}{\alpha|\mathcal{U}|}\sum_{i\in\mathcal{U}}\left[z_i^{(t+1)}\cdot\ell(\mathbf{V}^{(t)}\mathbf{u}_i, \mathcal{V}_i)\right] + \frac{1}{2}\|\mathbf{\Lambda}_u^{1/2}\mathbf{U}\|_F^2\right\}, \tag{41}$$

$$\mathbf{V}^{(t+1)} = \arg\min_{\mathbf{V}}\left\{\frac{1}{\alpha|\mathcal{U}|}\sum_{i\in\mathcal{U}}\left[z_i^{(t+1)}\cdot\ell(\mathbf{V}\mathbf{u}_i^{(t+1)}, \mathcal{V}_i)\right] + \frac{1}{2}\|\mathbf{\Lambda}_v^{1/2}\mathbf{V}\|_F^2\right\}. \tag{42}$$

Below, we present an efficient implementation of each step. The overall algorithm is described in Algorithm 3, including the update formulae for $\mathbf{U}$ and $\mathbf{V}$.

### D.1   Convolution-type smoothed quantile estimation

**Newton–Raphson algorithm.** The subproblem of $\xi$ for the general kernel density function $k_h$ cannot be solved analytically. Hence, we resort to a numerical solution using the efficient Newton–

Raphson (NR) method. At the $l$-th iteration in the $(t+1)$-th update, we estimate $\xi$ as follows:

$$\xi_{l+1}^{(t+1)} = \xi_l^{(t+1)} - \gamma_l \cdot d_l^{(t+1)}, \quad \text{where } d_l^{(t+1)} = \frac{\nabla_\xi \Psi_{1-\alpha}(\mathbf{U}^{(t)}, \mathbf{V}^{(t)}, \xi)}{\nabla_\xi^2 \Psi_{1-\alpha}(\mathbf{U}^{(t)}, \mathbf{V}^{(t)}, \xi)}. \tag{43}$$

Here, the first and second derivatives of $(1-\alpha)$-CtS-CVaR can be evaluated as follows:

$$\nabla_\xi \Psi_{1-\alpha}(\mathbf{U}, \mathbf{V}, \xi) = -\frac{1}{\alpha |\mathcal{U}|} \sum_{i \in \mathcal{U}} [(1-\alpha) - K_h(\xi - \ell(\mathbf{V}\mathbf{u}_i, \mathcal{V}_i))], \tag{44}$$

$$\nabla_\xi^2 \Psi_{1-\alpha}(\mathbf{U}, \mathbf{V}, \xi) = \frac{1}{\alpha |\mathcal{U}|} \sum_{i \in \mathcal{U}} k_h(\xi - \ell(\mathbf{V}\mathbf{u}_i, \mathcal{V}_i)). \tag{45}$$

Pre-computing the loss $\ell(\mathbf{V}\mathbf{u}_i, \mathcal{V}_i)$ for each user can be helpful in reducing the computational burden when the gradient and Hessian can be computed exactly. Furthermore, an effective initialization of $\xi$ is crucial, and we set $\xi_0^{t+1}$ to the previous estimate, i.e., $\xi_0^{t+1} = \xi_L^t$.

**Backtracking line search.** The value of $\gamma$ plays a vital role in obtaining an accurate solution for $\xi$. However, a constant $\gamma$ often fails to provide efficient results. One way to overcome this is by employing the widely-used backtracking line search to determine $\xi$ adaptively. It aims to find the maximum $\gamma \in \{1/2, 1/4, \dots\}$ such that the Armijo (sufficient decrease) condition (Armijo, 1966) is satisfied. We can express this as:

$$\gamma^* = \max_{\gamma \in \{1/2, 1/4, \dots\}} \gamma, \tag{46}$$
$$\text{s.t. } \Psi_{1-\alpha}(\mathbf{U}, \mathbf{V}, \xi - \gamma \cdot d) \le \Psi_{1-\alpha}(\mathbf{U}, \mathbf{V}, \xi) - c\gamma d \cdot \nabla_\xi \Psi_{1-\alpha}(\mathbf{U}, \mathbf{V}, \xi - \gamma \cdot d),$$

where $c > 0$ is the error tolerance parameter, often set to a small value such as $10^{-4}$. Performing each iteration of backtracking line search for evaluating $\Psi_{1-\alpha}(\mathbf{U}, \mathbf{V}, \xi - \gamma \cdot d)$ takes the cost of $\mathcal{O}(|\mathcal{U}|)$. Therefore, this step demands a cost of $\mathcal{O}(|\mathcal{U}|L)$ for each epoch.

**Sub-sampled algorithm.** Although the computation of the direction $d$ can be done in parallel for users, it may be costly for many users, particularly when using a backtracking line search with the cost of $\mathcal{O}(|\mathcal{U}|L)$. We can introduce the sub-sampled Newton–Raphson method (Roosta-Khorasani and Mahoney, 2019) to reduce this cost by approximating the gradient and Hessian based on the uniformly sub-sampled users. Let $|\mathcal{U}_b|$ be the sub-sample size of users, which is smaller than the original users size $|\mathcal{U}|$. Chernozhukov and Fernández-Val (2005) obtain the large-sample properties of the quantile regression estimator based on sub-sampling, under the setting where $|\mathcal{U}_b|/|\mathcal{U}| \to 0$ and $|\mathcal{U}_b| \to \infty$ as $|\mathcal{U}| \to \infty$. Their results suggest that the sub-sample estimator $\hat{\xi}_b$ of the true value $\xi_0$ satisfies that $\hat{\xi}_b = \xi_0 + O_p(1/\sqrt{|\mathcal{U}_b|})$. That is, the sub-sample estimator converges to the true parameter at a rate of $1/\sqrt{|\mathcal{U}_b|}$. In practice, the user size is often larger than ten million or more, and the sub-sample size $|\mathcal{U}_b| = 100,000$ ensures that the estimation error asymptotically vanishes as $1/\sqrt{|\mathcal{U}_b|} \approx 0.00316$.

### D.2 Re-weighted ALS

The update of $\mathbf{U}$ and $\mathbf{V}$ can be reformulated by using primal-dual splitting as in Eq. (11). Here, we focus on the update of primal variables, i.e., $\mathbf{U}$ and $\mathbf{V}$, since we have already described the $\mathbf{z}$ step in Section 3.3,

**U step.** Given $\mathbf{z}$, the optimization problem of $\mathbf{U}$ and $\mathbf{V}$ forms a re-weighted ERM. Owing to separability, the update of $\mathbf{U}$,

$$\mathbf{U}^{(t+1)} = \arg\min_{\mathbf{U}} \left\{ \frac{1}{\alpha |\mathcal{U}|} \sum_{i \in \mathcal{U}} [z_i^{(t+1)} \cdot \ell(\mathbf{V}^{(t)}\mathbf{u}_i, \mathcal{V}_i)] + \frac{1}{2} \|\mathbf{\Lambda}_u^{1/2}\mathbf{U}\|_F^2 \right\}, \tag{47}$$

can be solved in parallel with respect to each row $\mathbf{u}_i$ as follows:

$$\frac{z_i^{(t+1)}}{\alpha|\mathcal{U}|}\nabla_{\mathbf{u}_i}\ell(\mathbf{V}^{(t)}\mathbf{u}_i, \mathcal{V}_i) + \lambda_u^{(i)}\mathbf{u}_i = 0$$

$$\Longleftrightarrow \left(\underbrace{\frac{z_i^{(t+1)}}{|\mathcal{V}_i|}\sum_{j\in\mathcal{V}_i}\mathbf{v}_j^{(t)}\otimes\mathbf{v}_j^{(t)}}_{(a)} + \underbrace{z_i^{(t+1)}\beta_0\sum_{j\in\mathcal{V}}\mathbf{v}_j^{(t)}\otimes\mathbf{v}_j^{(t)}}_{(b)} + \alpha|\mathcal{U}|\lambda_u^{(i)}\mathbf{I}\right)\mathbf{u}_i = \frac{z_i^{(t+1)}}{|\mathcal{V}_i|}\sum_{j\in\mathcal{V}_i}\mathbf{v}_j^{(t)}.$$

$$(48)$$

The updated $\mathbf{u}_i$ can be obtained by solving the linear system above. Since the user-independent Gram matrix indicated by (b) has been pre-computed (at a cost of $\mathcal{O}(|\mathcal{V}|d^2)$), the computational cost of updating each user's $\mathbf{u}_i$ involves computing the user-dependent partial Hessian indicated by (a) in $\mathcal{O}(|\mathcal{V}_i|d^2)$ and then solving a linear system of $d\times d$ in $\mathcal{O}(d^3)$. Since $\sum_{i\in\mathcal{U}}|\mathcal{V}_i| = |\mathcal{S}|$, the total computational cost is thus $\mathcal{O}(|\mathcal{S}|d^2 + |\mathcal{U}|d^3)$.

**V step.** Analogously, the update of $\mathbf{V}$,

$$\mathbf{V}^{(t+1)} = \arg\min_{\mathbf{V}}\left\{\frac{1}{\alpha|\mathcal{U}|}\sum_{i\in\mathcal{U}}[z_i^{(t+1)}\cdot\ell(\mathbf{V}\mathbf{u}_i^{(t+1)}, \mathcal{V}_i)] + \frac{1}{2}\|\lambda_v^{1/2}\mathbf{V}\|_F^2\right\},\tag{49}$$

can be solved as follows:

$$\nabla_{\mathbf{v}_j}\frac{1}{\alpha|\mathcal{U}|}\sum_{i\in\mathcal{U}}z_i\cdot\ell(\mathbf{V}\mathbf{u}_i, \mathcal{V}_i) + \lambda_v^{(j)}\mathbf{v}_j = 0$$

$$\Longleftrightarrow \left(\sum_{i\in\mathcal{U}_j}\frac{z_i^{(t+1)}}{|\mathcal{V}_i|}\mathbf{u}_i^{(t+1)}\otimes\mathbf{u}_i^{(t+1)} + \underbrace{\beta_0\sum_{i\in\mathcal{U}}z_i^{(t+1)}\mathbf{u}_i^{(t+1)}\otimes\mathbf{u}_i^{(t+1)}}_{(c)} + \alpha|\mathcal{U}|\lambda_v^{(j)}\mathbf{I}\right)\mathbf{v}_j = \sum_{i\in\mathcal{U}_j}\frac{z_i^{(t+1)}}{|\mathcal{V}_i|}\mathbf{u}_i^{(t+1)}.$$

$$(50)$$

We can reduce the computational cost of this step as in the $\mathbf{U}$ step by caching the weighted Gram matrix indicated by (c), which costs $\mathcal{O}(|\mathcal{U}|d^2)$ when the loss for each user is pre-computed. The computational cost for updating $\mathbf{V}$ is $\mathcal{O}(|\mathcal{S}|d^2 + |\mathcal{V}|d^3)$.

### D.3 INSTANTIATION OF SAFER$_2$

An instance of SAFER$_2$ is determined by the choice of the kernel density function $k_h$. We shall describe the implementation of SAFER$_2$ with some popular kernels as it would be helpful for reproducing our method; the implementation described here is available in https://github.com/riktor/safer2-recommender.

**Gaussian kernel.** For the Gaussian kernel, the kernel density $k_h$ and its CDF $K_h$ can be computed as follows:

$$k_h(u) = \frac{1}{\sqrt{2\pi}h}\exp\left(-\frac{u^2}{2h^2}\right),\tag{51}$$

$$K_h(u) = \frac{1}{2}\left[1 + \text{erf}\left(\frac{u}{\sqrt{2}h}\right)\right] = \frac{1}{2}\text{erfc}\left(-\frac{u}{\sqrt{2}h}\right),\tag{52}$$

where $\text{erf}(u) = \frac{2}{\sqrt{\pi}}\int_0^u\exp(-v^2)dv$ and $\text{erfc}(u) = 1 - \text{erf}(u)$ are the error and complementary error functions, respectively.

These complex functions are generally implemented as special functions and can be computed very efficiently (Abrarov and Quine, 2011; Gautschi, 1970; Poppe and Wijers, 1990; Zaghloul and Ali, 2012).

The smoothed check function $(\rho_{1-\alpha} * k_h)$ is then obtained as

$$(\rho_{1-\alpha} * k_h)(u) = \frac{h}{2}\left[h\cdot k_h(u) + \frac{u}{h}(1 - 2\cdot K_h(-u))\right] + ((1-\alpha) - 0.5)u,\tag{53}$$

which is used for backtracking line-search in the $\xi$ step.

**Epanechnikov kernel.** We also describe the implementation for the Epanechnikov kernel (Epanechnikov, 1969). Epanechnikov kernel density $k_h$ and its CDF $K_h$ can be computed as follows:

$$k_h(u) = \frac{3}{4h}\left[1 - \left(\frac{u}{h}\right)^2\right]\mathbb{1}\{|u/h| \le 1\}, \tag{54}$$

$$K_h(u) = \begin{cases} 0, & u < -1 \\ \frac{1}{4h^3}\left[u(3h^2 - u^2) + 2h^3\right], & u/h \in [-1, 1] \\ 1, & u/h > 1 \end{cases}. \tag{55}$$

The smoothed check function $(\rho_{1-\alpha} * k_h)$ is then obtained as

$$(\rho_{1-\alpha} * k_h)(u) = \frac{h}{2}\left[\frac{3}{4}\left(\frac{u}{h}\right)^2 - \frac{1}{8}\left(\frac{u}{h}\right)^4 + \frac{3}{8}\right]\mathbb{1}\{|u/h| \le 1\}$$
$$+ \frac{h}{2}|u/h|\mathbb{1}\{u > 1\} + ((1-\alpha) - 0.5)u. \tag{56}$$

Note that the support of $k_h(u)$ is on $(-h, h)$, and thus we can ensure the strict convexity (i.e., positive Hessian) of $(\rho_{1-\alpha} * k_h)$ in a Newton–Raphson step.

### D.4 SAFER$_2$++

The SAFER$_2$ algorithm experiences the quadratic/cubic runtime dependency on the embedding size $d$. This problem has been tackled by various studies (Bayer et al., 2017; He et al., 2016; Pilászy et al., 2010), and some studies recently reported that the large dimension is very important to improve ranking quality of iALS (Ohsaka and Togashi, 2023; Rendle et al., 2021) . Considering this, we propose an extension of SAFER$_2$ for a large embedding size by using the recent subspace-based block coordinate descent of Rendle et al. (2021), which is a simple yet effective approach, enabling efficient utilization of optimized vector processing units.

**Subspace-based block coordinate descent.** To overcome the above problem, iALS++ considers the subvector of a user/item embedding as a block (Rendle et al., 2021) and optimizes the subvector by a Newton–Raphson method. We here apply this approach to our SAFER$_2$. Let $\boldsymbol{\pi} \subseteq \{1, \ldots, d\}$ be a vector of indices and $\mathbf{u}_{i,\boldsymbol{\pi}}$ be the subvector of $\mathbf{u}_i$ corresponding to $\boldsymbol{\pi}$. Then, the first and second derivatives of the objective $L_i(\mathbf{u}_i, \mathbf{V}) = (z_i/\alpha|\mathcal{U}|) \cdot \ell(\mathbf{V}\mathbf{u}_i, \mathcal{V}_i) + \frac{\lambda_u^{(i)}}{2}\|\mathbf{u}_i\|_2^2$ in Eq. (11) with respect to $\mathbf{u}_{i,\boldsymbol{\pi}}$ are:

$$\nabla_{\mathbf{u}_{i,\boldsymbol{\pi}}} L_i(\mathbf{u}_i, \mathbf{V}) \propto \frac{z_i}{|\mathcal{V}_i|}\sum_{j \in \mathcal{V}_i}(\mathbf{v}_j^\top \mathbf{u}_i - 1)\mathbf{u}_{i,\boldsymbol{\pi}} + z_i\beta_0\left(\sum_{j \in \mathcal{V}}\mathbf{v}_{j,\boldsymbol{\pi}}\mathbf{v}_j^\top\right)\mathbf{u}_i + \alpha|\mathcal{U}|\lambda_u^{(i)}\mathbf{u}_{i,\boldsymbol{\pi}}, \tag{57}$$

$$\nabla_{\mathbf{u}_{i,\boldsymbol{\pi}}}^2 L_i(\mathbf{u}_i, \mathbf{V}) \propto \frac{z_i}{|\mathcal{V}_i|}\sum_{j \in \mathcal{V}_i}\mathbf{v}_{j,\boldsymbol{\pi}}\mathbf{v}_{j,\boldsymbol{\pi}}^\top + z_i\beta_0\sum_{j \in \mathcal{V}}\mathbf{v}_{j,\boldsymbol{\pi}}\mathbf{v}_{j,\boldsymbol{\pi}}^\top + \alpha|\mathcal{U}|\lambda_u^{(i)}\mathbf{I}. \tag{58}$$

Note that we omitted the constant factor $(\alpha|\mathcal{U}|)^{-1}$ for brevity. We pre-compute the partial Gram matrices $\sum_{j \in \mathcal{V}}\mathbf{v}_{j,\boldsymbol{\pi}}\mathbf{v}_j^\top$ in $\mathcal{O}(|\mathcal{V}||\boldsymbol{\pi}|d)$, $\sum_{j \in \mathcal{V}}\mathbf{v}_{j,\boldsymbol{\pi}}\mathbf{v}_{j,\boldsymbol{\pi}}^\top$ in $\mathcal{O}(|\mathcal{V}||\boldsymbol{\pi}|^2)$, and the prediction $\mathbf{v}_j^\top \mathbf{u}_i$ for $(i, j) \in \mathcal{S}$ in $\mathcal{O}(|\mathcal{S}|d)$. Then, the computational cost of the gradient and Hessian for all users is $\mathcal{O}(|\mathcal{S}||\boldsymbol{\pi}| + |\mathcal{U}||\boldsymbol{\pi}|d + |\mathcal{S}||\boldsymbol{\pi}|^2)$. We subsequently update the subvector $\mathbf{u}_{i,\boldsymbol{\pi}}$ by a Newton–Raphson step,

$$\mathbf{u}_{i,\boldsymbol{\pi}}^{(t+1)} = \mathbf{u}_{i,\boldsymbol{\pi}}^{(t)} - (\nabla_{\mathbf{u}_{i,\boldsymbol{\pi}}}^2 L_i(\mathbf{u}_i^{(t)}, \mathbf{V}^{(t)}))^{-1}\nabla_{\mathbf{u}_{i,\boldsymbol{\pi}}} L_i(\mathbf{u}_i^{(t)}, \mathbf{V}^{(t)}). \tag{59}$$

This can be computed by solving a linear system of size $|\boldsymbol{\pi}| \times |\boldsymbol{\pi}|$ in $\mathcal{O}(|\boldsymbol{\pi}|^3)$ time. The computational cost for updating all user subvectors is $\mathcal{O}(|\mathcal{S}||\boldsymbol{\pi}| + |\mathcal{V}||\boldsymbol{\pi}|d + |\mathcal{S}||\boldsymbol{\pi}|^2 + |\mathcal{U}||\boldsymbol{\pi}|^3)$.

Similarly, denoting $L_j(\mathbf{U}, \mathbf{V}) = (\alpha|\mathcal{U}|)^{-1}\sum_{i \in \mathcal{U}}[z_i \cdot \ell(\mathbf{V}\mathbf{u}_i, \mathcal{V}_i)] + \frac{\lambda_v^{(j)}}{2}\|\mathbf{v}_j\|_2^2$, we can obtain the update of an item embedding as follows:

$$\mathbf{v}_{j,\boldsymbol{\pi}}^{(t+1)} = \mathbf{v}_{j,\boldsymbol{\pi}}^{(t)} - (\nabla_{\mathbf{v}_{j,\boldsymbol{\pi}}}^2 L_j(\mathbf{U}, \mathbf{V})^{-1}\nabla_{\mathbf{v}_{j,\boldsymbol{\pi}}} L_i(\mathbf{U}, \mathbf{V}), \tag{60}$$

---

**Algorithm 4:** SAFER$_2$++ solver.

---

$\forall i \in \mathcal{U}, \ \mathbf{u}_i \sim \mathcal{N}(\vec{0}, (\sigma/\sqrt{d})\mathbf{I}_d), \quad \forall j \in \mathcal{V}, \ \mathbf{v}_j \sim \mathcal{N}(\vec{0}, (\sigma/\sqrt{d})\mathbf{I}_d)$

$\xi \leftarrow 0$

**for** $t \leftarrow 1$ **to** $T$ **do**

    $\mathbf{G}_V \leftarrow \mathbf{V}^\top \mathbf{V} = \sum_{j \in \mathcal{V}} \mathbf{v}_j \mathbf{v}_j^\top$

    Compute $\forall i \in \mathcal{U}, \ \ell_i = \frac{1}{2|\mathcal{V}_i|} \sum_{j \in \mathcal{V}_i} (1 - \mathbf{u}_i^\top \mathbf{v}_j)^2 + \frac{\beta_0}{2} \cdot \mathbf{u}_i^\top \mathbf{G}_V \mathbf{u}_i$

    **for** $l \leftarrow 1$ **to** $L$ **do**

        Uniformly draw $\mathcal{U}_b \subseteq \mathcal{U}$

        $\widehat{d} \leftarrow \frac{\nabla_\xi \widehat{\Psi}_{1-\alpha}(\mathbf{U}, \mathbf{V}, \xi)}{\nabla_\xi^2 \widehat{\Psi}_{1-\alpha}(\mathbf{U}, \mathbf{V}, \xi)}$

        $\gamma \leftarrow \arg\max_{\gamma \in (0,1)} \gamma$

            s.t. $\widehat{\Psi}_{1-\alpha}(\mathbf{U}, \mathbf{V}, \xi - \gamma\widehat{d}) \leq \widehat{\Psi}_{1-\alpha}(\mathbf{U}, \mathbf{V}, \xi) - c\gamma\widehat{d}\nabla_\xi \widehat{\Psi}_{1-\alpha}(\mathbf{U}, \mathbf{V}, \xi - \gamma\widehat{d})$

        $\xi \leftarrow \xi - \gamma \cdot \widehat{d}$

    **end**

    **for** $i \leftarrow 1$ **to** $|\mathcal{U}|$ **do**

        $z_i \leftarrow 1 - K_h(\xi - \ell_i)$

    **end**

    **for** $\pi \in \mathcal{P}$ **do**

        $\mathbf{G}_{V,\pi}^{gl} \leftarrow \sum_{j \in \mathcal{V}} \mathbf{v}_{j,\pi} \mathbf{v}_j^\top, \ \mathbf{G}_{V,\pi}^{ll} \leftarrow \sum_{j \in \mathcal{V}} \mathbf{v}_{j,\pi} \mathbf{v}_{j,\pi}^\top$

        **for** $i \leftarrow 1$ **to** $|\mathcal{U}|$ **do**

            $\mathbf{u}_{i,\pi} \leftarrow \mathbf{u}_i - (\nabla_{\mathbf{u}_{i,\pi}}^2 L_i(\mathbf{u}_i, \mathbf{V}))^{-1} \nabla_{\mathbf{u}_{i,\pi}} L_i(\mathbf{u}_i, \mathbf{V})$

        **end**

        $\widetilde{\mathbf{G}}_{U,\pi}^{gl} \leftarrow \sum_{i \in \mathcal{U}} z_i \cdot \mathbf{u}_{i,\pi} \mathbf{u}_i^\top, \ \widetilde{\mathbf{G}}_{U,\pi}^{ll} \leftarrow \sum_{i \in \mathcal{U}} z_i \cdot \mathbf{u}_{i,\pi} \mathbf{u}_{i,\pi}^\top$

        **for** $j \leftarrow 1$ **to** $|\mathcal{V}|$ **do**

            $\mathbf{v}_{j,\pi} \leftarrow \mathbf{v}_{j,\pi} - (\nabla_{\mathbf{v}_{j,\pi}}^2 L_j(\mathbf{U}, \mathbf{V}))^{-1} \nabla_{\mathbf{v}_{j,\pi}} L_j(\mathbf{U}, \mathbf{V})$

        **end**

    **end**

**end**

---

where

$$\nabla_{\mathbf{v}_{j,\pi}} L_j(\mathbf{U}, \mathbf{V}) \propto \sum_{i \in \mathcal{U}_i} \frac{z_i}{|\mathcal{V}_i|} (\mathbf{u}_i^\top \mathbf{v}_j - 1)\mathbf{v}_{j,\pi} + \beta_0 \left( \sum_{i \in \mathcal{U}} z_i \cdot \mathbf{u}_{i,\pi} \mathbf{u}_i^\top \right) \mathbf{v}_j + \alpha|\mathcal{U}|\lambda_v^{(j)} \mathbf{v}_{j,\pi}, \quad (61)$$

$$\nabla_{\mathbf{v}_{j,\pi}}^2 L_j(\mathbf{U}, \mathbf{V}) \propto \sum_{i \in \mathcal{U}_j} \frac{z_i}{|\mathcal{V}_i|} \mathbf{u}_{i,\pi} \mathbf{u}_{i,\pi}^\top + \beta_0 \sum_{i \in \mathcal{U}} z_i \cdot \mathbf{u}_{i,\pi} \mathbf{u}_{i,\pi}^\top + \alpha|\mathcal{U}|\lambda_v^{(j)} \mathbf{I}. \quad (62)$$

The computational cost for updating all item subvectors is $\mathcal{O}(|\mathcal{S}||\pi| + |\mathcal{U}||\pi|d + |\mathcal{S}||\pi|^2 + |\mathcal{V}||\pi|^3)$.

**Computational complexity.** We follow the iteration scheme suggested by Rendle et al. (2021), which cyclically updates the subspace of user and item sides for each subset of indices. As a result, we obtain the following computational cost

$$\mathcal{O}\left( |\mathcal{S}|d + \frac{d}{|\pi|}(|\mathcal{U}| + |\mathcal{V}|)(d|\pi| + |\pi|^2 + |\pi|^3) + |\mathcal{S}|(|\pi| + |\pi|^2)) \right) \quad (63)$$

$$\equiv \mathcal{O}(|\mathcal{S}||\pi|d + (|\mathcal{U}| + |\mathcal{V}|)(d^2 + d|\pi|^2)). \quad (64)$$

The algorithm is shown in Algorithm 4.

## E ON TIKHONOV REGULARIZATION

In this appendix, we develop a regularization strategy for SAFER$_2$, which allows us to control the numerical stability of subproblems for users and items. Since setting appropriate regularization

weight for every user/item is impractical in large-scale settings, we derive a single hyperparameter that controls the regularization weights for all users and items.

For a matrix $\mathbf{A} \in \mathbb{R}^{d \times d}$, the condition number $\kappa(\mathbf{A})$ is defined as follows:

$$\kappa(\mathbf{A}) := \|\mathbf{A}^{-1}\|_F \cdot \|\mathbf{A}\|_F. \tag{65}$$

The condition number of a matrix $\mathbf{A}$ characterises the numerical stability of a linear system $\mathbf{A}\mathbf{x} = \mathbf{b}$; this problem with respect to $\mathbf{x}$ is numerically unstable when $\kappa(\mathbf{A})$ is large. We consider normal $\mathbf{A}$, and then the condition number can be computed as follows:

$$\kappa(\mathbf{A}) = \frac{|\lambda_{\max}(\mathbf{A})|}{|\lambda_{\min}(\mathbf{A})|}, \tag{66}$$

where $\lambda_{\max}(\mathbf{A})$ and $\lambda_{\min}(\mathbf{A})$ are maximal and minimal eigenvalues of $\mathbf{A}$, respectively. In our case, we want to keep small the condition number of each Hessian matrix with respect to the rows of $\mathbf{U}$ and $\mathbf{V}$. For the $i$-th row of $\mathbf{U}$, the Hessian matrix $\mathbf{H}_i$ is as follows:

$$\mathbf{H}_i := \frac{z_i}{|\mathcal{V}_i|} \sum_{j \in \mathcal{V}_i} \mathbf{v}_j \mathbf{v}_j^\top + z_i \beta_0 \mathbf{V}^\top \mathbf{V} + \alpha |\mathcal{U}| \lambda_u^{(i)} \mathbf{I}, \tag{67}$$

which implies

$$\lambda_{\min}(\mathbf{H}_i) = \alpha |\mathcal{U}| \lambda_u^{(i)}, \quad \lambda_{\max}(\mathbf{H}_i) = \lambda_{\max}(\mathbf{H}_i - \alpha |\mathcal{U}| \lambda_u^{(i)} \mathbf{I}) + \alpha |\mathcal{U}| \lambda_u^{(i)}. \tag{68}$$

To ensure a small value of $\kappa(\mathbf{H}_i)$, we can adjust the regularization weight $\lambda_u^{(i)} > 0$. However, the maximal eigenvalue $\lambda_{\max}(\mathbf{H}_i)$ changes at each update step in the alternating optimization, but computing the dominant eigenvalue is a costly process (Mises and Pollaczek-Geiringer, 1929). Therefore, we propose a simple regularization strategy to control the condition number of each linear system solved with a constant hyperparameter. Assuming $\nu > 0$ is the upper bound of the squared norm of the user and item embeddings throughout the optimization, i.e., $\nu \geq \|\mathbf{v}_j\|_2^2$ for all $j \in \mathcal{V}$ and $\nu \geq \|\mathbf{u}_i\|_2^2$ for all $i \in \mathcal{U}$, we have the following upper bound for $\lambda_{\max}(\mathbf{H}_i - \alpha |\mathcal{U}| \lambda_u^{(i)} \mathbf{I})$:

$$\begin{aligned}
\lambda_{\max}(\mathbf{H}_i - \alpha |\mathcal{U}| \lambda_u^{(i)} \mathbf{I}) &\leq \left\| \frac{z_i}{|\mathcal{V}_i|} \sum_{j \in \mathcal{V}_i} \mathbf{v}_j \mathbf{v}_j^\top + z_i \beta_0 \mathbf{V}^\top \mathbf{V} \right\|_F \\
&\leq \frac{z_i}{|\mathcal{V}_i|} \sum_{j \in \mathcal{V}_i} \|\mathbf{v}_j\|_2^2 + z_i \beta_0 \|\mathbf{V}\|_F^2 \\
&\leq z_i \nu + z_i \beta_0 |\mathcal{V}| \nu \\
&\leq \nu(1 + \beta_0 |\mathcal{V}|).
\end{aligned}$$

In the last inequality, we used $z_i \leq 1$. Therefore, by setting

$$\lambda_u^{(i)} = \frac{\lambda}{\alpha |\mathcal{U}|} (1 + \beta_0 |\mathcal{V}|) \tag{69}$$

with a hyperparameter $\lambda > 0$, we can ensure the following inequality,

$$\begin{aligned}
\kappa(\mathbf{H}_i) &= \frac{\lambda_{\max}(\mathbf{H}_i - \alpha |\mathcal{U}| \lambda_u^{(i)} \mathbf{I}) + \lambda_{\min}(\mathbf{H}_i)}{\lambda_{\min}(\mathbf{H}_i)} \\
&\leq \frac{\nu}{\lambda} + 1.
\end{aligned} \tag{70}$$

The advantage of this reparametrization is that we may be able to bound the condition number of each user's linear system from above by $(\nu/\lambda) + 1$, which is independent of $\alpha$, $|\mathcal{U}|$, $|\mathcal{V}_i|$ and $i$. This allows us to flexibly control the regularization intensity through tuning $\lambda$ while ensuring that the subproblems of each user are conditioned to the same extent. Note that a too-large value of $\lambda$ leads to poor model training while the condition number will be close to one, and therefore, we still need to tune $\lambda$.

Analogously, we can derive the regularization weight for the $j$-th row of $\mathbf{V}$.

$$\mathbf{H}_j := \sum_{i \in \mathcal{U}_j} \frac{z_i}{|\mathcal{V}_i|} \cdot \mathbf{u}_i \mathbf{u}_i^\top + \beta_0 \mathbf{U}^\top \mathrm{diagMat}(\mathbf{z}) \mathbf{U} + \alpha |\mathcal{U}| \lambda_v^{(j)} \mathbf{I}, \tag{71}$$

and

$$\lambda_{\max}(\mathbf{H}_j - \alpha|\mathcal{U}|\lambda_v^{(j)}\mathbf{I}) \leq \left\|\sum_{i\in\mathcal{U}_j} \frac{z_i}{|\mathcal{V}_i|}\mathbf{u}_i\mathbf{u}_i^\top + \beta_0\mathbf{U}^\top\text{diagMat}(\mathbf{z})\mathbf{U}\right\|_F$$

$$\leq \sum_{i\in\mathcal{U}_j} \frac{z_i}{|\mathcal{V}_i|}\|\mathbf{u}_i\|_2^2 + \beta_0 \cdot \sum_{i\in\mathcal{U}} z_i\|\mathbf{u}_i\|_2^2$$

$$\leq \nu \cdot \sum_{i\in\mathcal{U}_j} \frac{z_i}{|\mathcal{V}_i|} + \beta_0\nu \cdot \sum_{i\in\mathcal{U}} z_i$$

$$\leq \nu \left(\sum_{i\in\mathcal{U}_j} \frac{1}{|\mathcal{V}_i|} + \beta_0 \sum_{i\in\mathcal{U}} z_i\right).$$

In contrast to the case of user embeddings, we applied $z_i \leq 1$ for the first term of the last inequality. This is because bounding $\sum_{i\in\mathcal{U}} z_i \leq |\mathcal{U}|$ is rather loose, and the weighting strategy based on this bound will lead to the over-regularization of item embeddings. To avoid this, we introduce the following property of convolution-type smoothed quantile.

**Proposition E.1.** *Suppose that $n$ samples $\{\ell_1, \ldots, \ell_n\}$ of losses and its convolution-type smoothed quantile $\xi_n := \arg\min_\xi \sum_{i=1}^n (\rho_{1-\alpha} * k_h)(\ell_i - \xi)$. Then, $n$ dual variables $\{z_1, \ldots, z_n\}$ satisfy*

$$\sum_{i=1}^n z_i = \alpha \cdot n, \tag{72}$$

*where $z_i = 1 - K_h(\xi_n - \ell_i)$.*

*Proof.* The smoothed quantile $\xi_n$ satisfies the first-order optimality condition,

$$\nabla_\xi \sum_{i=1}^n (\rho_{1-\alpha} * k_h)(\ell_i - \xi_n) = 0 \iff \sum_{i=1}^n [(1-\alpha) - K_h(\xi_n - \ell_i)] = 0$$

$$\iff \sum_{i=1}^n [1 - K_h(\xi_n - \ell_i)] = \alpha \cdot n,$$

which immediately completes the proof. $\qquad\square$

From this result, we can substitute $\sum_{i\in\mathcal{U}} z_i$ in the upper bound of $\lambda_{\max}(\mathbf{H}_j - \alpha|\mathcal{U}|\lambda_v^{(j)}\mathbf{I})$ with $\alpha|\mathcal{U}|$ and then obtain

$$\lambda_{\max}(\mathbf{H}_j - \alpha|\mathcal{U}|\lambda_v^{(j)}\mathbf{I}) \leq \nu \left(\sum_{i\in\mathcal{U}_j} \frac{1}{|\mathcal{V}_i|} + \beta_0\alpha|\mathcal{U}|\right). \tag{73}$$

By setting

$$\lambda_v^{(j)} = \frac{\lambda}{\alpha|\mathcal{U}|}\left(\sum_{i\in\mathcal{U}_j} \frac{1}{|\mathcal{V}_i|} + \beta_0\alpha|\mathcal{U}|\right), \tag{74}$$

we can ensure the following inequality

$$\kappa(\mathbf{H}_j) = \frac{\lambda_{\max}(\mathbf{H}_j - \alpha|\mathcal{U}|\lambda_v^{(j)}\mathbf{I}) + \lambda_{\min}(\mathbf{H}_j)}{\lambda_{\min}(\mathbf{H}_j)}$$

$$\leq \frac{\nu(\sum_{i\in\mathcal{U}_j} \frac{1}{|\mathcal{V}_i|} + \beta_0\alpha|\mathcal{U}|)}{\alpha|\mathcal{U}|\lambda_v^{(j)}} + 1$$

$$= \frac{\nu(\sum_{i\in\mathcal{U}_j} \frac{1}{|\mathcal{V}_i|} + \beta_0\alpha|\mathcal{U}|)}{\lambda(\sum_{i\in\mathcal{U}_j} \frac{1}{|\mathcal{V}_i|} + \beta_0\alpha|\mathcal{U}|)} + 1$$

$$= \frac{\nu}{\lambda} + 1.$$

**On the regularization strategy of iALS.** Tikhonov regularization has been widely adopted for MF models with the ALS solver (Rendle et al., 2022; Zhou et al., 2008). In particular, the recent technique in Eq. (23) (proposed by Rendle et al. (2022)) can be obtained by a similar derivation. Namely, consider the Hessian matrix and regularization weight for the $i$-th user,

$$\mathbf{H}_i = \sum_{j \in \mathcal{V}_i} \mathbf{v}_j \mathbf{v}_j^\top + \beta_0 \mathbf{V}^\top \mathbf{V} + \lambda_u^{(i)} \mathbf{I}, \tag{75}$$

$$\lambda_u^{(i)} = \lambda \left( |\mathcal{V}_i| + \beta_0 |\mathcal{V}| \right), \tag{76}$$

then we have

$$\lambda_{\max}(\mathbf{H}_i - \alpha |\mathcal{U}| \lambda_u^{(i)} \mathbf{I}) \leq \| \sum_{j \in \mathcal{V}_i} \mathbf{v}_j \mathbf{v}_j^\top + \beta_0 \mathbf{V}^\top \mathbf{V} \|_F$$

$$\leq \sum_{j \in \mathcal{V}_i} \|\mathbf{v}_j\|_2^2 + \beta_0 \|\mathbf{V}\|_F^2 \|_F$$

$$\leq \nu(|\mathcal{V}_i| + \beta_0 |\mathcal{V}|),$$

which implies $\kappa(\mathbf{H}_i) \leq \nu/\lambda + 1$. The result for each item $j$ is analogous, and we therefore omit the derivation.

## F    ALTERNATING DIRECTION METHOD OF MULTIPLIERS (ADMM)

As discussed in Section 3, the smoothed check function $(\rho_1 * k_h)$ is non-linear and leads to the coupling between the rows of $\mathbf{V}$ as follows:

$$\min_{\mathbf{U}, \mathbf{V}, \xi} \left\{ \xi + \frac{1}{\alpha |\mathcal{U}|} \sum_{i \in \mathcal{U}} (\rho_1 * k_h) \left( \ell(\mathbf{V} \mathbf{u}_i, \mathcal{V}_i) - \xi \right) + \frac{1}{2} \|\mathbf{\Lambda}_u^{1/2} \mathbf{U}\|_F^2 + \frac{1}{2} \|\mathbf{\Lambda}_v^{1/2} \mathbf{V}\|_F^2 \right\}.$$

To decouple the rows of $\mathbf{V}$, one can consider the use of the alternating direction method of multipliers (ADMM) (Boyd et al., 2011; Steck et al., 2020; Togashi and Abe, 2022) by introducing auxiliary variables $\mathbf{y} \in \mathbb{R}^{|\mathcal{U}|}$, which leads to the following constrained optimization.

$$\min_{\mathbf{U}, \mathbf{V}, \xi} \left\{ \xi + \frac{1}{\alpha |\mathcal{U}|} \sum_{i \in \mathcal{U}} (\rho_1 * k_h) \left( y_i - \xi \right) + \frac{1}{2} \|\mathbf{\Lambda}_u^{1/2} \mathbf{U}\|_F^2 + \frac{1}{2} \|\mathbf{\Lambda}_v^{1/2} \mathbf{V}\|_F^2 \right\}, \tag{77}$$

$$\text{s.t. } y_i = \ell(\mathbf{V} \mathbf{u}_i, \mathcal{V}_i), \ \forall i \in \mathcal{U}.$$

The augmented Lagrangian in a scaled form is defined as follows:

$$L_\rho(\mathbf{U}, \mathbf{V}, \xi, \mathbf{y}, \mathbf{w}) = \xi + \frac{1}{\alpha |\mathcal{U}|} \sum_{i \in \mathcal{U}} (\rho_1 * k_h) \left( y_i - \xi \right) + \frac{1}{2} \|\mathbf{\Lambda}_u^{1/2} \mathbf{U}\|_F^2 + \frac{1}{2} \|\mathbf{\Lambda}_v^{1/2} \mathbf{V}\|_F^2$$

$$+ \frac{\rho}{2} \sum_{i \in \mathcal{U}} (w_i - \ell(\mathbf{V} \mathbf{u}_i, \mathcal{V}_i) + y_i)^2, \tag{78}$$

where $\mathbf{w} \in \mathbb{R}^{|\mathcal{U}|}$ is the dual variables (i.e., the Lagrange multipliers). Observe that, because of the quadratic penalty term of ADMM, the rows of $\mathbf{V}$ are still coupling in the objective. One can avoid this by using another reformulation by introducing $|\mathcal{V}|$-dimensional auxiliary variable $\mathbf{y}_i = \mathbf{V} \mathbf{u}_i$ for each user, which requires prohibitively large dual variables of size $|\mathcal{U}||\mathcal{V}|$.

## G    DETAILS AND ADDITIONAL RESULTS OF EXPERIMENTS

This appendix provides detailed descriptions of the experimental settings and additional results, which are omitted for the strict space limitation.

### G.1    MODELS

**iALS.** Our implementation of iALS is based on the reference software publicly provided by Rendle et al. (2022). This iALS implementation is reported to be competitive with state-of-the-art methods on *ML-20M* and *MSD* datasets. We used their proposed regularization strategy with $\nu = 1.0$ as suggested by Rendle et al. (2021). We also followed their implementation of iALS++ and set $\nu = 1.0$ for all settings as in iALS.

Table 3: Ranges of hyperparameters.

| Models | Hyperparameters |
|---|---|
| iALS | $\beta_0 \in [1e-2, 1.0], \lambda \in [5e-4, 0.1]$ |
| Mult-VAE | $\tau \in [1e-4, 1e-2], |\mathcal{U}_b| \in [50, 100, 200, 300, 400, 500], \beta \in [0.1, 1.0]$ |
| ERM-MF | $\beta_0 \in [1e-4, 1e-2], \lambda \in [1e-4, 1e-2]$ |
| CVaR-MF | $\beta_0 \in [1e-3, 0.1], \lambda \in [1e-4, 1e-2], \alpha = 0.3, \tau \in [0.1, 0.5]$ |
| SAFER$_2$ | $\beta_0 \in [1e-4, 1e-2], \lambda \in [1e-4, 1e-2], \alpha = 0.3, h \in [0.1, 1.0], |\mathcal{U}_b|/|\mathcal{U}| = 0.1, L = 5$ |

**ERM-MF and SAFER$_2$.** We implemented ERM-MF and SAFER$_2$ (SAFER$_2$++) in the same code-base as iALS. For ERM-MF and SAFER$_2$, we used our proposed Tikhonov regularization as we found it is generally effective in terms of the final quality and hyperparameter sensitivity.

**CVaR-MF.** The implementation of CVaR-MF is also provided in our software. As CVaR-MF often takes a much longer time to converge, we tune a constant learning rate $\tau > 0$ for $\mathbf{U}$ and $\mathbf{V}$. We also found that applying the gradient descent to the $\xi$ step is quite unstable and makes it hard to obtain acceptable performance. Therefore, we exactly compute the $(1 - \alpha)$-quantile of the users' loss as $\xi$ in each step. To finely observe the difference of the solvers, we used our proposed regularization also for CVaR-MF, which empirically leads to good performance.

**Mult-VAE.** We implemented the method based on the public codebase.[8] We tune a learning rate $\tau > 0$, batch size $|\mathcal{U}_b|$, and annealing parameter $\beta > 0$.

**Prediction for new users.** As we follow the strong generalization setting, each MF-based model must produce predictions for new users who are not in the training split and thus do not have the trained embeddings (e.g., $\mathbf{u}_i$). To this end, we follow previous studies (e.g., Rendle et al. (2022)), where each model solves an independent convex problem for each user by leveraging the 80% of the user's interactions. In iALS, we can obtain the embedding $\mathbf{u}_i$ of a new user $i$ with $\mathcal{V}_i$ by solving the following problem:

$$\mathbf{u}_i = \underset{\mathbf{u} \in \mathbb{R}^d}{\arg\min} \left\{ \ell_{\text{iALS}}(\mathbf{Vu}, \mathcal{V}_i) + \frac{\lambda_u^{(i)}}{2} \|\mathbf{u}\|_2^2 \right\}, \tag{79}$$

where $\lambda_u^{(i)} = \lambda(|\mathcal{V}_i| + \beta_0 |\mathcal{V}|)$.

In ERM-MF, CVaR-MF, and SAFER$_2$, the problem can be expressed as follows:

$$\mathbf{u}_i = \underset{\mathbf{u} \in \mathbb{R}^d}{\arg\min} \left\{ \frac{1}{\alpha |\mathcal{U}|} \ell(\mathbf{Vu}, \mathcal{V}_i) + \frac{\lambda_u^{(i)}}{2} \|\mathbf{u}\|_2^2 \right\}, \tag{80}$$

where we set $\lambda_u^{(i)} = (\lambda/\alpha|\mathcal{U}|)(1 + \beta_0 |\mathcal{V}|)$, and $\mathbf{V}$ is the trained item embedding matrix; we fix it in the prediction phase. This is a standard least-square problem and hence easy to solve by computing the analytical solution. Note that applying this prediction procedure even to the gradient-based CVaR-MF is reasonable because the subproblems for users are completely independent in this step.

**Hyperparameter tuning.** In the experiments in Section 5, we tuned all models by using the validation split for each dataset. The range of each hyperparameter is presented in Table 3. We tuned all the hyperparameters by performing a grid search for *ML-1M*. For *ML-20M* and *MSD*, we tuned all the parameters manually to reduce the experimental burden. The number of epochs $T$ is set to 20 for iALS, ERM-MF, and SAFER$_2$ in the hyperparameter search and set to 50 for training the final models. For CVaR-MF, we set $T = 500$ for validation and $T = 1,000$ for testing.

### G.2 EVALUATION PROTOCOL

**Datasets and pre-processing protocol.** We employed a standard pre-processing protocol for the datasets (Liang et al., 2018; Rendle et al., 2022; Steck, 2019b; Weimer et al., 2007). The

---

[8]https://github.com/younggyoseo/vae-cf-pytorch

implementation of the pre-processing protocol is based on Liang et al. (2018). As we described in Section 5, we divided the users into three subsets: the training subset (i.e., $\{\mathcal{V}_i\}_{i \in \mathcal{U}}$) contains 80% of the users, and the remaining users are split into two holdout subsets for validation and testing purposes; For each validation and testing subset of **ML-1M**, **ML-20M**, and **MSD**, the number of users evaluated is 1,000, 10,000, and 50,000, respectively.

**Evaluation measures.** In our experiments, we use recall at $K$ (R@$K$) and normalized discounted cumulative gain at $K$ (nDCG@$K$) as measures of ranking quality. Let $\mathcal{V}'_i \subset \mathcal{V}$ be the held-out items pertaining to user $i$, and $\pi_i(k) \in \mathcal{V}$ be the $k$-th item on the ranked list evaluated for user $i$. The computation of R@$K$ and DCG@$K$ follow:

$$\text{R@}K(\pi_i, \mathcal{V}'_i) = \frac{1}{\min(K, |\mathcal{V}'_i|)} \sum_{k=1}^{K} \mathbb{1}\{\pi_i(k) \in \mathcal{V}'_i\}, \tag{81}$$

$$\text{DCG@}K(\pi_i, \mathcal{V}'_i) = \sum_{k=1}^{K} \frac{\mathbb{1}\{\pi_i(k) \in \mathcal{V}'_i\}}{\log_2(k+1)}. \tag{82}$$

nDCG@$K$ is defined as $\text{nDCG@}K(\pi_i, \mathcal{V}') = \text{DCG@}K(i, \pi_i)/\text{DCG@}K(i, \pi_i^*)$ where $\pi_i^*$ is an ideal ranking for user $i$.

### G.3 ADDITIONAL EXPERIMENTS

**Effect of the sub-sampled Newton-Raphson method.** Figure 7 shows the effect of the number of sub-samples $|\mathcal{U}_b|$ for each sub-sampled NR iteration in the $\xi$ step. Each curve was obtained by varying $|\mathcal{U}_b|$ with the best hyperparameter setting for $|\mathcal{U}_b|/|\mathcal{U}| = 0.1$. We can observe that (1) the primal variables (i.e., $\mathbf{U}$ and $\mathbf{V}$) converge even with a small $|\mathcal{U}_b|$; (2) the dual variables (i.e., $\mathbf{z}$) fluctuate with small $|\mathcal{U}_b|$ values; and (3) the final ranking qualities are almost identical for different $|\mathcal{U}_b|$ values. It suggests that the sub-sampled NR method is effective in practice as it alleviates the computational cost of the $\xi$ step, which is the only additional cost from iALS.

We also report the effect of the number of iterations $L$ in the $\xi$ step in Figure 8. There is a similar trend in Figure 7: Small values of $L$ and $|\mathcal{U}_b|$ lead to the fluctuation of $\mathbf{z}$, whereas the convergence is maintained in most cases. The final quality slightly deteriorates when both $L$ and $|\mathcal{U}_b|$ are small, particularly in terms of the semi-worst-case performance (i.e., $\alpha = 0.3$).

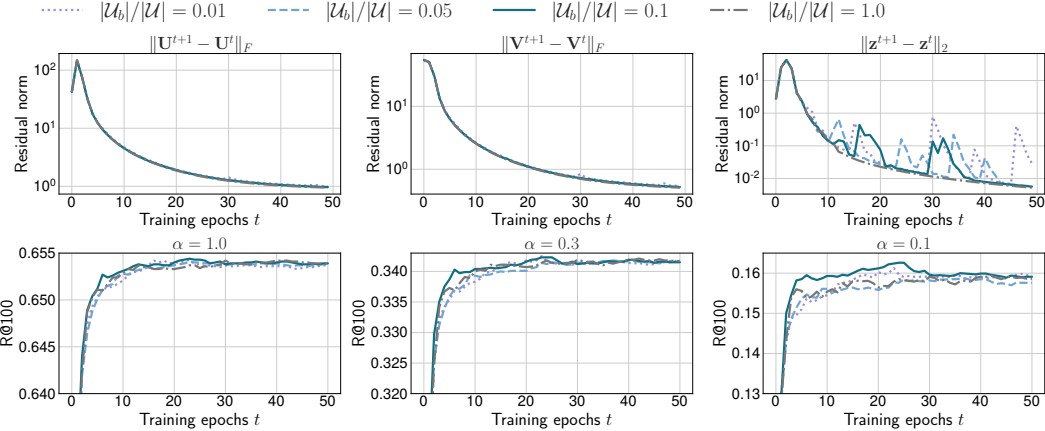

Figure 7: Convergence profile of SAFER$_2$ with different sampling ratio $|\mathcal{U}_b|/|\mathcal{U}|$ on **ML-20M**.

**Choice of kernels.** Various symmetric kernels can be used to instantiate SAFER$_2$ as discussed in Appendix D.3. To observe the effect of choosing kernel functions, we compare SAFER$_2$ with the Gaussian kernel, as examined in Section 5, and with the Epanechnikov kernel. Figure 9 demonstrates the distribution of relative ranking performance of each method compared to iALS through nested cross-validation as in Section 5. The SAFER$_2$ instance with the Gaussian kernel performs slightly

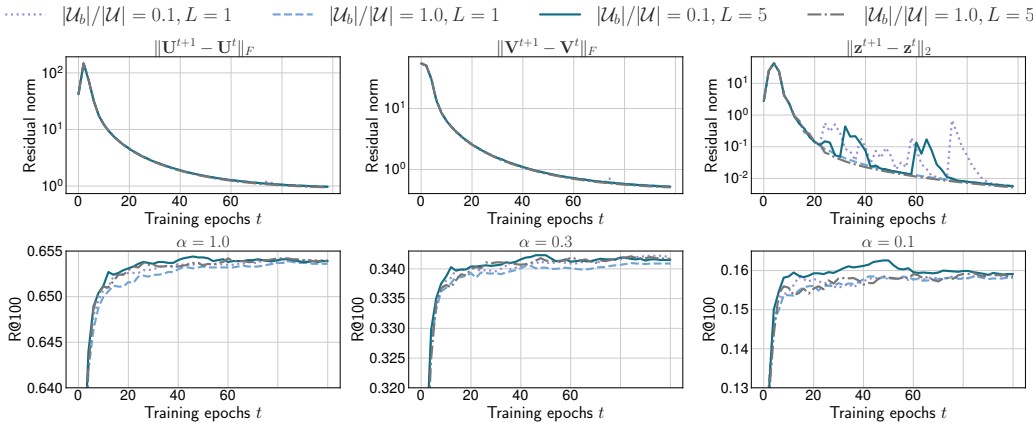

Figure 8: Convergence profile of SAFER$_2$ with different number of NR iterations $L$ on **ML-20M**.

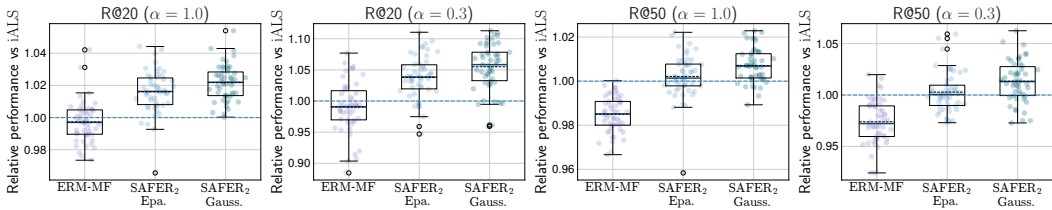

Figure 9: Distributions of relative performance vs iALS on **ML-1M**.

better than the one with the Epanechnikov kernel, but the difference between them is not substantial. However, the Gaussian kernel results in stability in the $\xi$ step due to its full support property, making it easier to tune the bandwidth parameter $h$.

**Inexact linear solvers for large embedding size.** Because SAFER$_2$ has quadratic/cubic runtime dependency on the embedding size $d$, we proposed a variant of SAFER$_2$, i.e., SAFER$_2$++, in Appendix D.4. Here, we examine the computational efficiency of SAFER$_2$++. To establish a baseline method for comparison, we consider another variant of SAFER$_2$, called SAFER$_2$-CG, that uses the conjugate gradient (CG) method to solve $d \times d$ linear systems. For SAFER$_2$-CG, we used the CG implementation in the Eigen library[9]; the maximum number of iterations was set to five, and the error tolerance was set to $1\mathrm{e}{-4}$. We used the same hyperparameters for all models, which achieved the best ranking quality with the exact linear solver and $d = 256$ for the validation split of **ML-20M**. In Figure 10, we present the convergence speed of SAFER$_2$++ on **ML-20M** for different values of $d$ and $|\pi|$. For comparison, we also display the red dashed curve that corresponds to the original SAFER$_2$'s results, except for the case of $d = 4{,}096$, where SAFER$_2$ did not finish in a practical time. Our results show that SAFER$_2$++ achieves comparable ranking quality compared to SAFER$_2$ for both average and semi-worst-case scenarios. Furthermore, although the convergence speed in terms of training epochs is similar for both SAFER$_2$ and SAFER$_2$++, SAFER$_2$++ exhibits substantially superior computational performance. When $d$ is small (e.g., $d = 256$ in the second row of the figure), SAFER$_2$-CG (light green line) outperforms SAFER$_2$++ in terms of wall time. However, the performance gain of SAFER$_2$++ increases for larger values of $d$, such as $d = 1{,}024$, $2{,}048$, or $4{,}096$, highlighting the scalability of SAFER$_2$++ with respect to the embedding size.

---

[9] https://eigen.tuxfamily.org

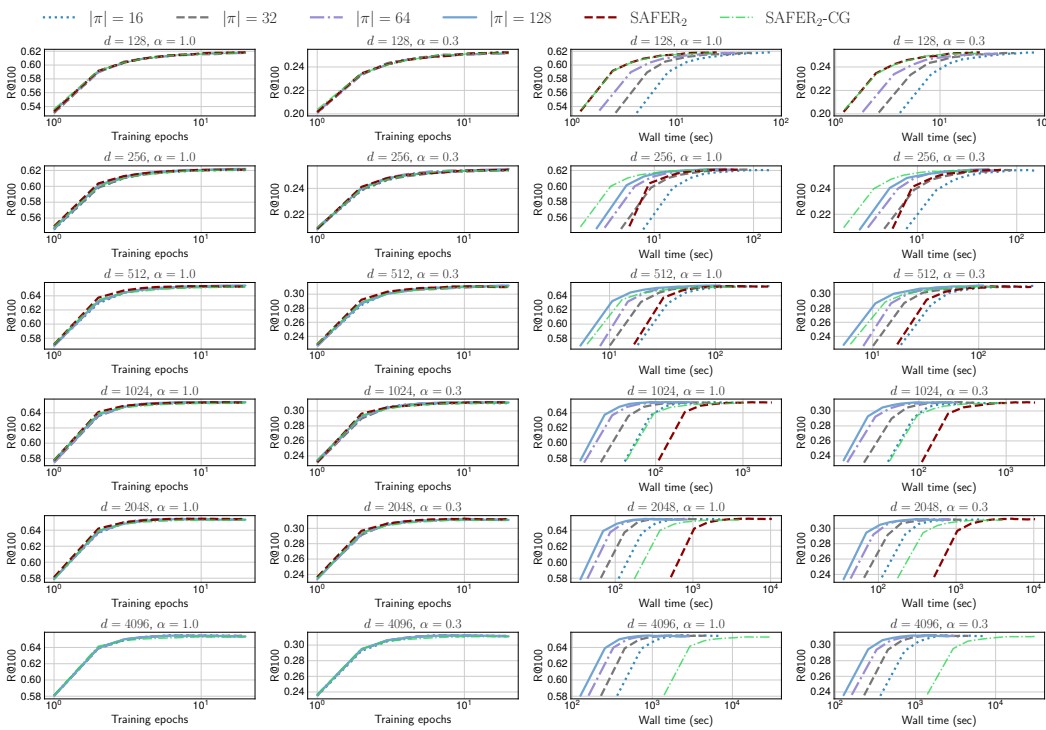

Figure 10: Effect of embedding and subspace block sizes on convergence speed of SAFER$_2$++.

