# OpenReview forum: "Safe Collaborative Filtering"
_ICLR.cc/2024/Conference — ICLR 2024 poster_

### Official Review · Reviewer_QW8m · 2023-11-01

**Soundness:** 4 excellent
**Presentation:** 3 good
**Contribution:** 4 excellent
**Rating:** 8
**Confidence:** 2

**Summary:**

The authors target tail performance in recommendation systems- a crucial but often overlooked scenario in the recommendation domain. They propose a collaborative filtering-based training approach for minimizing the conditional value at risk (CVaR), which represents the average risk for tail users. This is in contrast to conventional ERM-based methods, which focus on the overall average performance. The authors first show that a reformulation of the ERM objective using matrix factorization includes a non-smooth and non-linear term, which hampers the objective’s separability, making it difficult to parallelize and scale the approach. The authors circumvent this issue by applying convolution-type smoothing in the objective. They then use convolution-type smoothed quantile estimation in their proposed method, “Smoothing Approach for Efficient Risk-averse Recommender (SAFER2). They evaluate their model in the semi-worst case and average-case scenarios and find that, in general, their approach improves tail performance while maintaining average case performance.

**Strengths:**

1. The authors propose a novel and efficient approach for making CVaR minimization tractable, allowing them to target tail performance in their training objective.
2. Their approach improves tail performance without significantly impacting average performance.
3. Rigorous mathematical proofs are provided for justifying each step of their algorithm.
4. The authors compare the performance of their model with several baselines and state of the art methods.
5. The experiments also demonstrate that the proposed method is robust and computationally efficient.

**Weaknesses:**

Some concepts may need to be explained in more detail in the main paper (e.g.- the reformulation of the CVaR minimization objective, smoothing techniques for quantile regression, NR algorithm etc.)

**Questions:**

No particular questions.

---

> ### Author Response · Authors · 2023-11-16
> **Response to Reviewer QW8m**
>
> **The authors sincerely appreciate your valuable comments on enhancing the quality of our paper. We hope that we have addressed all your concerns in our response. If there are any points we missed or if you have any further questions, we would be delighted to continue the discussion.**
>
> First of all, we appreciate your recognition of the novel aspects of our CVaR minimization approach and the mathematical rigor underpinning our methodology. The emphasis you placed on our comprehensive comparative analysis and the robustness and efficiency of our method is particularly encouraging. These aspects underscore the practical importance of our research for large-scale recommender systems.
>
> Next, we also appreciate your suggestion to provide more detailed explanations of specific concepts in our main paper. Owing to page limitations, we directed these explanations to the Appendix in our Supplementary Material. In the revised version, we will clarify these concepts further, as you have suggested. We believe this revision will enhance the paper's readability and appeal to a broader audience. We greatly appreciate your valuable input.

---

### Official Review · Reviewer_RZsQ · 2023-11-01

**Soundness:** 4 excellent
**Presentation:** 3 good
**Contribution:** 3 good
**Rating:** 8
**Confidence:** 3

**Summary:**

Authors propose a method for recommender systems via Matrix Factorization, that is having a special type of loss-function that is connected to conditional value at risk (CVaR in Finance).
However, CVaR is "hard" for optimization, author re-use Rockafellar and Uryasev (2000) reformulation of CVAR to block multi-convex  loss function, over which CVaR-MF optimzation can be done. Still this formulation is not scalable, since max(0,x) part prevents separability for items. Author propose to use Molifier kernels to get strict convexity.
Finally, they wrap the steps in Smoothing Approach For Efficient Risk-averse Recommender (SAFER2).

**Strengths:**

Going beyond standard ERM objectives, nice mathematical re-formulation of tail risk CVaR and combining it with Matrix Factorization for Recommender Systems.

**Weaknesses:**

No clear weakness, except more baselines from Robust Recommender Systems field.

**Questions:**

Can you make comparison to the existing model:
e.g. Wen, Hongyi, et al. "Distributionally-robust Recommendations for Improving Worst-case User Experience." Proceedings of the ACM Web Conference 2022. 2022.

In your paper, major contributions are done by forcing re-formulation of CVaR to be scalable. Can you elaborate, what is the problem of controlling the "tail-risk" with distribution-wise losses like divergences (KL), or Wasserstein metrics?

---

> ### Author Response · Authors · 2023-11-15
> **Response to Reviewer RZsQ**
>
> **The authors sincerely appreciate your valuable comments on enhancing the quality of our paper. We hope that we have addressed all your concerns in our response. If there are any points we missed or if you have any further questions, we would be delighted to continue the discussion.**
>
> ## Questions
> 1. Can you make comparison to the existing model: e.g. Wen, Hongyi, et al. "Distributionally-robust Recommendations for Improving Worst-case User Experience." Proceedings of the ACM Web Conference 2022.
> 2. In your paper, major contributions are done by forcing re-formulation of CVaR to be scalable. Can you elaborate, what is the problem of controlling the "tail-risk" with distribution-wise losses like divergences (KL), or Wasserstein metrics?
>
> ## Response 1
> Wen et al. (2022) utilized the softmax cross-entropy loss and focused on minimizing the average loss by applying weighted averages for known subgroups. While their approach and ours are not mutually exclusive and could be seen as complementary, there are clear distinctions.
>
> Firstly, our methodology prioritizes reducing user losses at the extreme ends of the distribution (the tail) by solving a convex problem at each step, derived from a meticulously designed loss function combined with matrix factorization models. This strategy stabilizes the solution path and maintains the necessary convergence speed in practice. In contrast, Wen et al. (2022) aim to minimize average loss across groups, dynamically updating model parameters, group-wise loss estimates, and group weights via stochastic gradient descent/ascent. While their approach offers an interesting model architecture, it faces "a few challenges specific to recommendation" as noted in their Sections 3.2.2 and 4. Their method, similar to Mult-VAE in the use of neural networks and listwise loss functions, differs in the requirement to update loss estimates and weights for groups. As a result, their method is expected to demonstrate a slower convergence speed compared to Mult-VAE, and significantly slower than ours.
>
> Secondly, our method operates independently of any prior knowledge about subgroups. However, it is conceptually easy to integrate group-specific information if desired, making it compatible with the algorithmic structure proposed by Wen et al. (2022). It is important to note, though, that their method may introduce computational challenges as mentioned earlier, and combining our approach with theirs could be computationally demanding in practice.
>
> Third, our loss function is uniquely designed to address users at the extremities of the loss distribution, rather than centering on the average losses across subgroups. This approach offers planners a more refined and targeted control over the loss metrics they seek to minimize.
>
> ## Response 2
> The growing body of research on distributionally robust estimation often employs the "Min-Max" principle, where the focus is on minimizing the worst-case performance across a specific class of distributions. Research in this area often restricts the class of distributions using metrics such as KL divergence or Wasserstein metrics, aiming to minimize average loss while maintaining robustness against deviations within these predefined classes.
>
> Our method diverges from this norm by directly minimizing the Conditional Value at Risk (CVaR), thereby enhancing user satisfaction at the tail end of the distribution. This approach resonates with a fundamental principle in machine learning, eloquently stated by Vapnik: "When solving a problem of interest, do not solve a more general problem as an intermediate step. Try to get the answer that you really need but not a more general one" (Vapnik, 1995, 2006). In this vein, our direct approach is more fitting compared to methodologies that indirectly constrain CVaR through divergences or Wasserstein metrics. Nevertheless, it's important to acknowledge that a similar objective might be attainable through careful selection of distribution classes, albeit requiring sophisticated knowledge and precise customization.
>
> When considering the relationship between CVaR and KL divergence, the potential role of Entropic Value at Risk (EVaR) as an intermediary is often brought to light. EVaR, deriving insights from KL divergence, is designed to tackle the computational challenges in estimating CVaR and also serves as an upper bound for it. Consequently, imposing constraints on EVaR implies a loose bounding of CVaR as well. However, our approach pivots away from this intermediary, focusing instead on directly computing CVaR. This direct methodology not only circumvents the need for EVaR but also proves scalable in large-scale applications such as recommendation systems.

---

> > ### Author Response · Authors · 2023-11-15
> > **References**
> >
> > ### References
> >
> > Vapnik, N. Vladimir. The nature of statistical learning theory. Springer Science & Business Media, 1995.
> >
> > Vapnik, N. Vladimir. Estimation of dependences based on empirical data. Springer Science & Business Media, 2006.
> >
> > Wen, H., Yi, X., Yao, T., Tang, J., Hong, L. and Chi, E.H., 2022, April. Distributionally-robust Recommendations for Improving Worst-case User Experience. In Proceedings of the ACM Web Conference 2022 (pp. 3606-3610).

---

> ### Comment · Reviewer_RZsQ · 2023-11-21
> **comment on reply**
>
> Thanks for your replies and quote from Vapnik "When solving a problem of interest, do not solve a more general problem as an intermediate step. Try to get the answer that you really need but not a more general one" (Vapnik, 1995, 2006)."
> My questions were aimed at positioning contributions in right way. I am ok with answers.
> I do not have anything new to add, except to try to add some of this clarifications to a paper.

---

> > ### Author Response · Authors · 2023-11-22
> > **Response to Reviewer RZsQ**
> >
> > We are grateful for your insightful questions and constructive feedback. Our discussion has been pivotal in clarifying the positioning of our paper within the existing body of literature. We will revise our paper to include the key points you have highlighted, ensuring that it more accurately mirrors the comprehensive nature of our discussions. Your contributions have undoubtedly played a significant role in enhancing both the content and the relevance of our work.

---

### Official Review · Reviewer_nPPr · 2023-11-09

**Soundness:** 3 good
**Presentation:** 3 good
**Contribution:** 3 good
**Rating:** 6
**Confidence:** 3

**Summary:**

The paper proposes SAFER2 for excellent tail performance. \
The paper modified CVaR and devises CVaR-MF and SAFER2.

**Strengths:**

1. Well-written and organized.
2. Solid derivation and equations.

**Weaknesses:**

[Major cencerns]
1. Motivation
- As far as I understand, the main motivation of this paper is tail performance. (in Abstract)
- However, I found that the main contribution of this paper focuses on parallel computation and Scalability. (in the manuscript.)
- What if we just put some high weight on the tail users? or What if we just compute gradient only with bottom-$\alpha$% losses?

2. Experiment
- The main experimental result focuses on that SAFER2 shows comparable results when $\alpha=1$ and superior results when $\alpha=0.3$.
- When $\alpha=1$, there are many recent methods outperforming Multi-VAE. e.g., LightGCN, SGCL. and in my opinion, the pair-wise and list-wise loss functions outperform the point-wise loss.
- When $\alpha=0.3$, there are many recent methods enhancing the performance of tail users.
- Did you train two SAFER2 models with $\alpha=1$ and $\alpha=0.3$? If so, that is somewhat unfair for the baselines (e.g., VAE-CF).

[Minor cencerns]
1. Equation numbers are needed for each equation when finding the location of the equation by ordering.

**Questions:**

Please refer to Weaknesses.

---

> ### Author Response · Authors · 2023-11-15
> **Response to Reviewer nPPr**
>
> **The authors sincerely appreciate your valuable comments on enhancing the quality of our paper. We hope that we have addressed all your concerns in our response. We would be delighted to continue the discussion if there are any points we missed or if you have any further questions.**
> > As far as I understand, the main motivation of this paper is tail performance (in Abstract). However, I found that the main contribution of this paper focuses on parallel computation and scalability (in the manuscript).
>
> The main focus of this work is tail performance optimization, with a particular emphasis on scalability. We believe that recommender systems are particularly critical applications of machine learning where scalability is paramount. Therefore, instead of applying existing CVaR optimization methods (e.g., CVaR-MF), we develop an approach to achieve scalability in large-scale recommender systems.
>
> > What if we just compute gradient only with bottom-$\alpha$% losses?
>
> We interpreted your comment as focusing on the bottom-$\alpha$% of users rather than individual user-item interactions.
> The suggested approach is essentially consistent with CVaR-MF, which focuses on the bottom-$\alpha$% users. Hence, **we believe this hypothesis has been thoroughly examined in our manuscript.**
>
> > What if we just put some high weight on the tail users?
>
> There are various weighting strategies for this approach. In this light, CVaR-MF can be viewed as a naive implementation that equally weights tail users while disregarding others. Our algorithm can also be viewed as one that assigns smooth weights to users,　represented by dual variables $\mathbf{z}$. This leads to an ALS-type algorithm with user reweighting (Section 3.3), which achieves excellent scalability.
> Therefore, **we believe our manuscript comprehensively examines both of the approaches you've suggested.**
>
> > Did you train two SAFER2 models with $\alpha=1.0$ and $\alpha=0.3$? If so, that is somewhat unfair for the baselines (e.g., VAE-CF).
>
> **We set $\alpha=0.3$ for both CVaR-MF and SAFER2 throughout all experiments.**
> Although we have already discussed this in Section 5.1 and Table 3 in the supplementary material, we acknowledge your suggestion and will improve the clarity of this point in the revised manuscript.
>
> > When $\alpha=1.0$, there are many recent methods outperforming Multi-VAE. e.g., LightGCN, SGCL.
>
> Our evaluation operates within the strong generalization setting. However, **the suggested GNN-based methods cannot be evaluated in this setting.** These models cannot make predictions for new users. In fact, we could find few references that compare such methods in a strong generalization setting; Mao et al. (2021) have attempted to compare their GCN-type method in this protocol with some modifications to their method (see Section 4.8 of Mao et al. (2021)).
>
> Finally, we would like to emphasize that we do not intend to claim that SAFER2 is a state-of-the-art method regarding the average ranking quality; in fact, we have reported the superior average performance of Mult-VAE on ML-20M in Table 1. We are confident our experiments sufficiently support our claim that SAFER2 improves tail performance while maintaining scalability and average ranking quality competitive with iALS, which is the fastest method with excellent ranking quality.
>
> > When $\alpha=1.0$, in my opinion, the pair-wise and list-wise loss functions outperform the point-wise loss.
>
> To our knowledge, **pairwise loss functions are not considered competitive in this setup, and they are not commonly evaluated as baselines**; for example, see Section 7 of Steck et al. (2020), Section 4.4 of Liang et al. (2018), and Table 3 of Rendle et al. (2022). Also noteworthy in this context is the conclusion of Rendle et al. (2022), *"It is striking that iALS achieves competitive or better performance than models learned with ranking losses (LambdaNet, WARP, softmax) which reflect the top-n recommendation task more closely."*  Furthermore, **we have already compared listwise loss functions.** The multinomial likelihood of Mult-VAE is essentially a listwise loss (a.k.a. softmax cross-entropy); see Eq. (2) of Liang et al. (2018).
> > When $\alpha=0.3$, there are many recent methods enhancing the performance of tail users.
>
> Thank you for your feedback. We would appreciate it if you could provide references for this.
>
> > Equation numbers are needed for each equation when finding the location of the equation by ordering.
>
> We will ensure the equation numbers are displayed as you suggested, and we appreciate your suggestion.
>
>
> ## References
> Mao, Kelong, et al. "SimpleX: A simple and strong baseline for collaborative filtering." CIKM 2021.
>
> Steck, Harald, et al. "Admm slim: Sparse recommendations for many users." WSDM 2020.
>
> Liang, Dawen, et al. "Variational autoencoders for collaborative filtering." WWW 2018.
>
> Rendle, Steffen, et al. "Revisiting the performance of ials on item recommendation benchmarks." RecSys 2022.

---

> ### Comment · Reviewer_nPPr · 2023-11-15
> **Response**
>
> Thank you for your thorough response. \
> I will raise my score by one step. \
> However, I cannot recommend this paper for acceptance.
>
> You mentioned that "we have reported the superior average performance of Mult-VAE on ML-20M in Table 1. We are confident our experiments sufficiently support our claim that SAFER2 improves tail performance while maintaining scalability".
> 1. Comparable performance with Multi-VAE is not sufficient since there are too many outperforming recent methods. Multi-VAE was proposed in 2018.
> 2. Enhancing recommendation performance for tail users (e.g., cold-start, popularity bias) is a widely studied concept in recent papers. However, the authors do not compare the tail performance with recent methods in the experiment.

---

> > ### Author Response · Authors · 2023-11-23
> > **Response to Reviewer nPPr: Part 2 [1/4]**
> >
> > **The authors sincerely appreciate your valuable comments on enhancing the quality of our paper. We hope that we have addressed all your concerns in our response.**
> >
> > First and foremost, we are grateful that you have taken our explanation into consideration and adjusted our marks accordingly. We would now like to provide further clarification regarding the remaining issues you kindly raised, offering explanations to affirm our stance on each point.
> >
> > > Comparable performance with Multi-VAE is not sufficient since there are too many outperforming recent methods. Multi-VAE was proposed in 2018.
> >
> > Thank you for your insightful suggestion. As mentioned in Section 2 of our manuscript, iALS has been reported to exhibit state-of-the-art comparable performance in the **2022** RecSys reproducible track (Rendle et al., 2022). The version of iALS that we have used for comparison is the latest version of this algorithm. On the other hand, we acknowledge that there are some methods that aim to improve ranking quality (rather than scalability) on ML-20M and MSD; e.g., Kim et al. (2019), Lobel et al. (2020), Shenbin et al. (2020). These methods, like Multi-VAE, are based on neural networks and experience slow convergence speeds due to SGD. Therefore, even if we were to include some of these methods in our experiments, it would not change our hypotheses or conclusions regarding the scalability/efficiency of our proposed method, particularly in comparison to neural network-based approaches. However, in response to the reviewer's insightful suggestions and in respect of their valuable time, we present an additional experiment here.
> >
> >
> > ## References
> > Kim, Daeryong, and Bongwon Suh. "Enhancing VAEs for collaborative filtering: flexible priors & gating mechanisms." RecSys 2019.
> >
> > Lobel, Sam, et al. "Towards amortized ranking-critical training for collaborative filtering." ICLR 2020.
> >
> > Shenbin, Ilya, et al. "Recvae: A new variational autoencoder for top-n recommendations with implicit feedback." WSDM 2020.
> >
> > Rendle, Steffen, et al. "Revisiting the performance of ials on item recommendation benchmarks." RecSys 2022.

---

> > > ### Author Response · Authors · 2023-11-23
> > > **Response to Reviewer nPPr: Part 2 [2/4]**
> > >
> > > ## Additional experiment
> > > We conduct a comparison with RecVAE proposed by Shenbin et al. (2020), which is one of the state-of-the-art VAE-based methods. In this experiment, we use the best hyperparameter settings for RecVAE suggested in the original paper. Note that RecVAE adopts an alternating strategy to optimize the user and item encoders, in which the user encoder is first updated using SGD for a number of iterations corresponding to 3 epochs, after which the item encoder is updated for one epoch. We refer to this series of updates, where each model parameter is updated at least once, as one epoch for RecVAE; this is in accordance with the original paper. We also consider the variant of our proposed algorithm, SAFER2++, which allows us to efficiently optimize SAFER2 with a large embedding size (described in Appendix D.4). We use the same hyperparameter setting for SAFER2 and SAFER2++, which is used in the main experiment of the current manuscript.
> > >
> > > ### Ranking quality vs training speed
> > > Tables A-D demonstrate the training speed of RecVAE and our methods in terms of ranking quality on ML-20M and MSD. Each column indicates the training wall time; "NA" indicates the end of model training. Similar to Mult-VAE, RecVAE's implementation utilizes an NVIDIA P100 GPU to accelerate model training. We can observe a similar trend as seen in Figure 5 of the current manuscript: While our methods do not outperform RecVAE on ML-20M, they show comparable or superior performance on MSD. It is remarkable that, for MSD, SAFER2 (with a small embedding size) still achieves competitive performance with RecVAE. Additionally, SAFER2++ clearly outperforms RecVAE in terms of both average and semi-worst-case quality on MSD.
> > >
> > > Table A: Convergence speed of RecVAE and SAFER2 (SAFER2++) in terms of R@100 on ML-20M.
> > > | Method | 20 sec  |  50 sec  |  100 sec |  200 sec |  500 sec | 1,000 sec |
> > > |--------|---------|----------|----------|----------|----------|----------|
> > > | RecVAE | 0.0447  |  0.5243  |  0.6214  |  0.6483  |  0.6686  |  0.6760  |
> > > | SAFER2 ($d=256$) | 0.0000  |  0.6495  |  NA      |   NA     |   NA     |  NA      |
> > > | SAFER2++ ($d=2{,}048$) | 0.0000  |  0.5739  |  0.6373  |  0.6494  |  0.6518  |  NA      |
> > >
> > >
> > > Table B: Convergence speed of RecVAE and SAFER2 (SAFER2++) in terms of R@100 ($\alpha=0.3$) on ML-20M.
> > > | Method | 20 sec  |  50 sec  |  100 sec |  200 sec |  500 sec | 1,000 sec |
> > > |--------|---------|----------|----------|----------|----------|----------|
> > > | RecVAE | 0.0000  |  0.1796  |  0.2918  |  0.3320  |  0.3557  |  0.3626  |
> > > | SAFER2 ($d=256$) | 0.0000  |  0.3344  |  NA  |  NA  |  NA  |  NA      |
> > > | SAFER2++ ($d=2{,}048$) | 0.0000  |  0.2346  |  0.315   |  0.3322  |  0.3359  |  NA      |
> > >
> > >
> > > Table C: Convergence speed of RecVAE and SAFER2 (SAFER2++) in terms of R@100 on MSD.
> > > | Method   | 200 sec |  500 sec |  1,000 sec|  2,000 sec|  5,000 sec| 10,000 sec|
> > > |----------|---------|----------|----------|----------|----------|----------|
> > > | RecVAE   | 0.0130  |  0.2912  |  0.3912  |  0.4330  |  0.4538  |  0.4583  |
> > > | SAFER2 ($d=512$) | 0.4552  |  0.4602  |  0.4600  |  NA      |  NA      |   NA     |
> > > | SAFER2++ ($d=2{,}048$)| 0.4461  |  0.4959  |  0.4980  |  0.4979  |  NA  |  NA  |
> > >
> > >
> > > Table D: Convergence speed of RecVAE and SAFER2 (SAFER2++) in terms of R@100 ($\alpha=0.3$) on MSD.
> > > | Method   | 200 sec |  500 sec |  1,000 sec|  2,000 sec|  5,000 sec| 10,000 sec|
> > > |----------|---------|----------|----------|----------|----------|----------|
> > > | RecVAE   | 0.0000  |  0.0465  |  0.1168  |  0.1488  |  0.1638  | 0.1672   |
> > > | SAFER2 ($d=512$) | 0.1634  |  0.1696  |  0.1700  |  NA      |   NA     |   NA     |
> > > | SAFER2++ ($d=2{,}048$)| 0.1507  |  0.1903  |  0.1922  |  0.1918  |  NA  | NA   |

---

> ### Author Response · Authors · 2023-11-23
> **Response to Reviewer nPPr: Part 2 [3/4]**
>
> ### Scalability w.r.t. the number of users and items
> We also examine the scalability of RecVAE and our methods.
> Due to RecVAE's alternating SGD algorithm mentioned above, it is not straightforward to compare RecVAE and our method in terms of the runtime per epoch.
> We hence discuss the relative training speed of each method on the two datasets (i.e., ML-20M and MSD), which are of different sizes; MSD has approximately 4 times the number of users and twice the number of items compared to ML-20M.
> As in the above experiment, the embedding size of SAFER2 is set to $d=256$ for ML-20M and $d=512$ for MSD. That of SAFER2++ is set to $d=2,048$ for both of the datasets.
>
>
> Table E shows the runtime per epoch of each method on ML-20M and MSD. Despite the SGD-based algorithm, which does not require the separability of the objective function, RecVAE is 9.6 times slower on MSD than on ML-20M. This is due to its $O(|\mathcal{U}||\mathcal{V}|)$ complexity for a single epoch, and therefore, RecVAE is not infeasible in real-world applications or requires some approximation techniques, which would lead to the quality degradation, e.g., sampled softmax (Bengio and Senécal, 2003). By contrast, SAFER2 is very efficient on both ML-20M and MSD, while it takes 16.7 times longer for MSD than ML-20M. The dominant factor in SAFER2's complexity is $\mathcal{O}(|\mathcal{U}|d^3)$, which stems from solving $d \times d$ linear systems for users; since the number of users is much larger than that of items on ML-20M and MSD, the cost for users becomes significant. More precisely, we solved each linear system by using the Cholesky decomposition with the approximated cost of $(1/3)d^3$. Therefore, since we set $d=256$ and $d=512$ for SAFER2, this result aligns with the theoretical complexity, $4 \cdot ((1/3) \cdot 2^3)=10.66... \approx 16.7$, which demonstrates the scalability of SAFER2 with respect to the numbers of users and items. It can also be observed in the result of SAFER2++: when the embedding size is fixed, SAFER2++ takes only 4 times longer for MSD than ML-20M. In summary, our proposed approach is highly scalable while maintaining average and tail performance, whereas RecVAE lacks scalability with respect to the number of users/items in terms of runtime per epoch, in addition to its slow convergence demonstrated in the previous experiment.
>
> Table E: Runtime/epoch on ML-20M and MSD.
> | Method               | ML-20M    | MSD | MSD / ML-20M |
> | --------             | -------- | -------- | -------- |
> | RecVAE               | 40.9 sec   | 395.0 sec | 9.6 |
> | SAFER2   | 3.4  sec    | 57.0 sec    | 16.7 |
> | SAFER2++ | 40.0 sec    | 166.2 sec  | 4.1 |
>
>
> #### Supplementary discussion
> SAFER2++ overcomes the cubic dependency on $d$ in SAFER2; comparing the relative runtime between SAFER2 and SAFER2++, SAFER2++ takes only $40.0 / 3.4 = 11.8 < (1/3)\cdot(2{,}048/256)^3=170.66...$ and $166.2 / 57.0=2.92 < (1/3)\cdot(2{,}048/512)^3=21.33...$ times longer for $d=2,048$ than $d=256,512$ on ML-20M and MSD, respectively.
>
>
> ## References
> Bengio, Yoshua, and Jean-Sébastien Senécal. "Quick training of probabilistic neural nets by importance sampling." AISTATS 2003.

---

> > ### Author Response · Authors · 2023-11-23
> > **Response to Reviewer nPPr: Part 2 [4/4]**
> >
> > > Enhancing recommendation performance for tail users (e.g., cold-start, popularity bias) is a widely studied concept in recent papers. However, the authors do not compare the tail performance with recent methods in the experiment.
> >
> >
> > We acknowledge the ongoing research developments addressing various challenges that could lead to unsatisfactory performance among tail-end users. For multiple reasons, we select state-of-the-art methods as our competitors.
> >
> > First, we would like to emphasize a key point of our paper: our approach is agnostic to the specific causes of inferior performance among tail users. As you pointed out, numerous factors, such as popularity bias and the cold-start problem, might contribute to this inferior performance at the tail. However, the effectiveness of these methods largely depends on whether their foundational mechanisms indeed underlie the observed problems. Therefore, comparing our method with existing ones necessitates a detailed, case-by-case analysis, which is beyond the scope of an article with limited space.
> >
> > Second, the objective of our approach differs from that of existing methods. These methods, tailored to specific situations, primarily focus on empirical risk minimization (ERM) to enhance average performance. They are often developed to tackle data sparsity issues such as popularity bias and cold-start. In contrast, our approach focuses on minimizing the Conditional Value at Risk (CVaR). This involves training the model to enhance the satisfaction of users who typically receive relatively poor recommendation results under the prediction model. This approach can play an important role even in scenarios where there are sufficient user-item interactions, and our algorithm maintains the practical computation speed necessary for large-scale recommendation systems.

---

> ### Comment · Reviewer_nPPr · 2023-11-23
> **Response**
>
> Thank you for your full response.\
> It is really impressive and appreciated.
>
> My last major concerns were
> 1. Comparable performance with Multi-VAE is not sufficient since there are too many outperforming recent methods. Multi-VAE was proposed in 2018.
> 2. Enhancing recommendation performance for tail users (e.g., cold-start, popularity bias) is a widely studied concept in recent papers. However, the authors do not compare the tail performance with recent methods in the experiment.
>
> For those,
> 1. I did not mean the SOTA VAE-based methods. However, as you already noted, there are tremendous recent methods that outperform VAE-CF. For example, you mentioned GCN-based cannot be evaluated in your setting, however, there are many inductive GCN-based CF methods. In the same manner, I can name at least five recent methods outperforming VAE-CF and able to be evaluated in your setting. The point is, even though you mentioned iALS has been reported to exhibit state-of-the-art comparable performance in the 2022 RecSys reproducible track (Rendle et al., 2022), that is one paper and there are many other papers outperforming iALS and VAE-CF.
> 2. Enhancing recommendation performance for tail users (e.g., cold-start, popularity bias) is a widely studied concept in recent papers. However, the authors do not compare the tail performance with recent methods in the experiment. In the same manner, there are many recent methods for tail users.
>
> For these reasons, I cannot give a strong positive score on this paper. \
> However, in light of your impressive response, I will raise my score to 6, which is on the positive side.

---

> > ### Author Response · Authors · 2023-11-23
> > **Response to Reviewer nPPr**
> >
> > **We sincerely appreciate your thoughtful consideration and constructive feedback. Your contributions have undoubtedly played a significant role in enhancing the content of our work.**
> >
> > > I did not mean the SOTA (State of the Art) VAE-based methods. However, as you already noted, there are many recent methods that outperform VAE-CF. For example, you mentioned that GCN-based methods cannot be evaluated in your setting, but there are many inductive GCN-based CF methods. Similarly, I can name at least five recent methods that outperform VAE-CF and can be evaluated in your setting. The point is, even though you mentioned that iALS has been reported to exhibit state-of-the-art comparable performance in the 2022 RecSys reproducible track (Rendle et al., 2022), that is one paper and there are many other papers that outperform iALS and VAE-CF.
> >
> > We acknowledge the comparison of our methods with inductive CF methods to be an intriguing question for discussion in our future work.
> > Since a pivotal aspect of our approach is ranking users based on their loss, we believe that the weak generalization/inductive setting would favor our approach, whereas we opted for the alternative setting to ensure a fair comparison. Thank you for your insightful discussion.
> >
> > > Enhancing recommendation performance for tail users (e.g., cold-start, popularity bias) is a widely studied concept in recent papers. However, the authors do not compare the tail performance with recent methods in the experiment. In the same manner, there are many recent methods for tail users.
> >
> > Our response in Part 2 [4/4] is the most comprehensive answer we can give at this point. We acknowledge that it is indeed an interesting topic in practice. As mentioned above, since the comparison results between such methods and ours depend on the model and data, in future work, we plan to conduct a more detailed investigation between such model/data-specific models. We are grateful for your suggestion of this exciting future direction for our study.

---

### Meta-Review · Area_Chair_KRPg · 2023-12-14

**Metareview:**

The paper introduces a safe collaborative filtering method that prioritizes recommendation quality for less-satisfied users by minimizing the conditional value at risk (CVaR). This approach addresses the challenge of tail performance in personalized recommender systems, ensuring effective handling of challenging samples within a dataset. The study develops an algorithm that extends the implicit alternating least squares method to overcome computational challenges for web-scale recommender systems. Empirical evaluation on real-world datasets demonstrates the excellent tail performance of the proposed approach while maintaining competitive computational efficiency.

The paper addresses the limitations of conventional methods based on empirical risk minimization and provides a solution that focuses on tail users' satisfaction and risk, thus enhancing the quality of personalized recommendations. "Robust Collaborative Filtering" fits the content better.

**Justification For Why Not Higher Score:**

The paper use the term "safe" but it is more focused on robustness and scalability. A detailed analysis of the algorithm's computational complexity would be helpful.

**Justification For Why Not Lower Score:**

Solid paper with reasonable improvements based on the existing CF work.

---

### Decision · Program_Chairs · 2024-01-16

Accept (poster)